# Earlier Tokens Contribute More: Learning Direct Preference Optimization From Temporal Decay Perspective

**Ruichen Shao**[1*]   **Bei Li**[1*†]   **Gangao Liu**[2,3]   **Yang Chen**[1]   **Xiang Zhou**[1]
**Jingang Wang**[1†]   **Xunliang Cai**[1]   **Peng Li**[2,3]
[1]Meituan Inc.   [2]Institute of Software, Chinese Academy of Sciences
[3]University of Chinese Academy of Sciences
{shaoruichen,libei17,chenyang108,wangjingang02}@meituan.com
{liugangao2023,lipeng}@iscas.ac.cn

## Abstract

Direct Preference Optimization (DPO) has gained attention as an efficient alternative to reinforcement learning from human feedback (RLHF) for aligning large language models (LLMs) with human preferences. Despite its advantages, DPO suffers from a length bias, generating responses longer than those from the reference model. Existing solutions like SimPO and SamPO address this issue but uniformly treat the contribution of rewards across sequences, overlooking temporal dynamics. To this end, we propose an enhanced preference optimization method that incorporates a temporal decay factor controlled by a gamma parameter. This dynamic weighting mechanism adjusts the influence of each reward based on its position in the sequence, prioritizing earlier tokens that are more critical for alignment. By adaptively focusing on more relevant feedback, our approach mitigates overfitting to less pertinent data and remains responsive to evolving human preferences. Experimental results on several benchmarks show that our approach consistently outperforms vanilla DPO by 5.9-8.8 points on AlpacaEval 2 and 3.3-9.7 points on Arena-Hard across different model architectures and sizes. Furthermore, additional experiments on mathematical and reasoning benchmarks (MMLU, GSM8K, and MATH) confirm that our method enhances performance without compromising general capabilities. Our codebase would be available at https://github.com/LotuSrc/D2PO.

## 1 Introduction

Direct Preference Optimization (DPO) (Rafailov et al., 2023) has recently emerged as a highly efficient alternative for aligning large language models (LLMs) with human preferences (Askell et al., 2021; Ouyang et al., 2022). Unlike reinforcement learning from human feedback (RLHF), which involves training a reward model followed by iterative policy updates, DPO reframes the problem as a binary classification task directly over human preference data. Compared to supervised fine-tuning, DPO enables the model not only to learn what is good but also to be aware of what is bad. This formulation allows DPO to optimize preference alignment in a single-stage training process, bypassing the complexities of reinforcement learning, such as policy sampling or extensive hyperparameter tuning. By leveraging an analytical mapping between reward functions and optimal policies, DPO fine-tunes LLMs efficiently and stably, offering superior performance in tasks like sentiment control, summarization, and dialogue generation while reducing computational overhead.

Despite its advantages, DPO suffers from a length bias problem, which is caused by the unbalanced length preference due to the non-uniform length distribution of chosen and rejected responses. This leads to generated responses tending to be longer than those of the reference model if the majority of chosen responses are longer than the rejected ones. To address this, SimPO (Meng et al., 2024)

---

*Equal Contributions.
†Corresponding Author.

introduces a more streamlined framework by eliminating the need for a reference model. Instead of relying on a pre-trained reference model for comparison, SimPO uses the average log probability of a generated sequence as the implicit reward signal. This innovation reduces computational complexity and memory usage, making SimPO a more efficient alternative to DPO. However, our experiments have revealed that SimPO suffers from unexpected performance issues when applied to data not generated through self-sampling. Similarly, SamPO (Lu et al., 2024) addresses DPO's length bias by limiting reward computation to the shorter time-series range between chosen and rejected responses, thereby the refining preference optimization.

Both of these studies, however, treat the contribution of each reward across the entire sequence as uniform. We posit that this uniform treatment cannot fully capture the nuances of preference optimization. Specifically, the temporal dynamics within a sequence may influence the importance of certain tokens or segments over others. To validate this conjecture, we plot the KL divergence between the instruct models and their DPO variants on three widely used open-source models, where the results are shown in Figure 1. We notice that the KL divergence remains larger at earlier tokens but gradually decreases along the positions, which indicates earlier tokens' distributions are more likely affected by DPO. This observation aligns with the findings of previous studies (Lin et al., 2024; Yang et al., 2023) that alignment is more critical for earlier to-

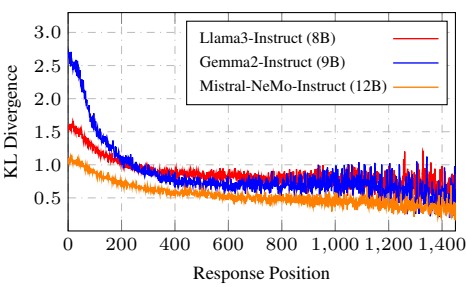

Figure 1: Visualization of KL divergence of instruct models and their DPO variants. The results include three widely used open-source LLMs: Llama3, Gemma2, and Mistral-NeMo. Observation here indicates earlier tokens contribute more during alignment.

kens. This is also consistent with the nature of next-token prediction, where an accurate prefix allows subsequent tokens to be generated on a more reliable foundation, thereby improving the overall quality (Edunov et al., 2018). In other words, the uncertainty of earlier tokens is much lower, and the calibration for more recent tokens is higher than that for earlier ones (Wang et al., 2020).

Building on this observation, we propose an enhanced version of DPO, namely temporal decay based DPO (short for $D^2PO$ ), that integrates a temporal decay factor, controlled by a gamma parameter, to further refine the influence of preference data during training. Our method introduces a dynamic weighting mechanism that modulates the contribution of each reward based on its temporal relevance, allowing the model to prioritize earlier feedback over more recent tokens. To this end, when the coefficient is slightly less than 1, it gradually reduces the influence of more recent rewards, which are inherently dependent on past rewards. Surprisingly, this temporal decay strategy has also been validated in the text-to-image synthesis task (Yang et al., 2024), where they emphasize the earlier steps in the reverse chain of the diffusion process. Our work complements theirs by showcasing the effectiveness of this approach in an autoregressive context, particularly in standard RLHF tasks. We provide a theoretical analysis on how earlier tokens can contribute more significantly from a token-level Markov Decision Process (MDP) perspective. Further discussion on the differences can be found in Appendix G.

By incorporating this adaptive temporal decay mechanism, $D^2PO$ not only facilitates earlier tokens to contribute more but also maintains the computational efficiency that makes DPO such a compelling approach for preference optimization. Experimental results on several widely used benchmarks, including AlpacaEval2, Arena-Hard and MT-bench, demonstrate the effectiveness on both off-policy and on-policy configurations. For example, in on-policy setups, $D^2PO$ outperforms DPO by up to 5.9-8.8 performance gains in terms of win rate on AlpacaEval2 and 3.3-9.7 points on Arena-Hard, respectively. Similarly, in off-policy setups, our method also demonstrates performance improvements. As a bonus of this decay mechanism which helps in reducing the overestimation of rewards caused by length bias in preference pairs, our method could be easily extended to reference-free mode, and it also can beat SimPO (Meng et al., 2024) by a large margin. Specifically, our best reference-free $D^2PO$ model can achieve 62.4 LC win rate on AlpacaEval 2 and 63.6 win rate on Arena Hard, which is competitive with the reference-based model. We also conducted additional experiments on mathematical and reasoning benchmarks, such as MMLU, GSM8K, and Math, indicating that our method enhances performance without compromising general capabilities.

## 2 RELATED WORK

### 2.1 REINFORCEMENT LEARNING FROM HUMAN FEEDBACK(RLHF)

The classical RLHF pipeline (Christiano et al., 2017; Ziegler et al., 2019; Ouyang et al., 2022) consists of two distinct stages: The reward modeling phase and the RL phase.

**Reward Modeling Phase.** The reward modeling is a binary classification task. Given a prompt, the comparison pair $(y_1, y_2)$ is collected by querying the supervised fine-tuning (SFT) model. Then, the preference $y_w \succ y_l$ is labeled by human which is used to train a reward model. Typically, Bradley-Terry model (Bradley & Terry, 1952) which quantifies the likelihood of one action being preferred over another is usually used to modeling the preference relations:

$$p\left(y_1 \succ y_2 \mid x\right) = \frac{\exp\left(r\left(x, y_1\right)\right)}{\exp\left(r\left(x, y_1\right)\right) + \exp\left(r\left(x, y_2\right)\right)} = \sigma\left(r\left(x, y_1\right) - r\left(x, y_2\right)\right) \quad (1)$$

**Reinforcement Learning Phase.** With the reward model in place, the second phase involves optimizing a policy through reinforcement learning, such as proximal policy optimization (PPO) (Schulman et al., 2017), aiming to maximize the learned reward while ensuring the policy remains close to a predefined reference policy (Korbak et al., 2022). This optimization is crucial for preventing model drift and maintaining alignment with human judgments, which is typically formulated as:

$$\max_\theta \mathbb{E}_{x \sim D, y \sim \pi_\theta(\cdot | x)}\left[r_\phi(x, y)\right] - \beta \mathbb{E}_{x \sim D}\left[\mathrm{KL}\left(\pi_\theta(\cdot \mid x) \| \pi_{\mathrm{ref}}(\cdot \mid x)\right)\right] \quad (2)$$

### 2.2 DIRECT ALIGNMENT ALGORITHMS (DAAS)

RLHF has become a cornerstone in the training of LLMs, facilitating their alignment with human preferences. However, the classical RLHF framework (Ouyang et al., 2022) is characterized by a two-stage training process, which includes reward modeling, and reinforcement learning. This complexity introduces several challenges and limitations, including reward over-optimization (Gao et al., 2022; Dubois et al., 2023; Wang et al., 2024a), training instability (Wu et al., 2023) and efficiency (Wang et al., 2024b). Nowadays, DAAs have emerged as a promising alternative, which can be divided into two major categories based on whether to consider a reference model.

**Reference-based Methods.** The reference-based methods in DAAs utilize a pre-existing model, often a supervised fine-tuned (SFT) model, as a reference point during the optimization process. This reference model serves as a baseline to which the updated model is compared, ensuring that updates do not deviate excessively from the initial, presumably safe and aligned, model configuration. DPO (Rafailov et al., 2023) is the most popular reference-based alignment algorithm and after its appearance, more researchers attempt to modify objective function for better performance. KTO (Ethayarajh et al., 2024) distinguishes itself by its capability to train from non-paired preference data, providing a unique angle on optimization. IPO (Azar et al., 2024) learns directly from preferences without relying on the Bradley-Terry model assumption that assumes that pairwise preferences can be substituted with pointwise rewards. R-DPO (Park et al., 2024) is an enhanced derivative of DPO, fortified with an additional regularization term designed to mitigate the tendency to exploit length biases, thus ensuring more balanced and diverse response generation.

**Reference-free Methods.** In contrast to reference-based methods that depend on a pre-existing model for guidance, reference-free methods forgo the need for such a reference. They directly optimize the model parameters in response to human feedback, which can enhance the flexibility of the optimization process. However, this freedom also presents challenges in controlling the extent of updates. CPO (Xu et al., 2024) leverages sequence likelihood as a reward signal and is trained in conjunction with an SFT objective. ORPO (Hong et al., 2024) is a novel alignment method that integrates an odds ratio-based penalty into the supervised fine-tuning process. SimPO (Meng et al., 2024) uses an average log probability as an implicit reward and introduces a target reward margin to enhance performance without relying on a reference model.

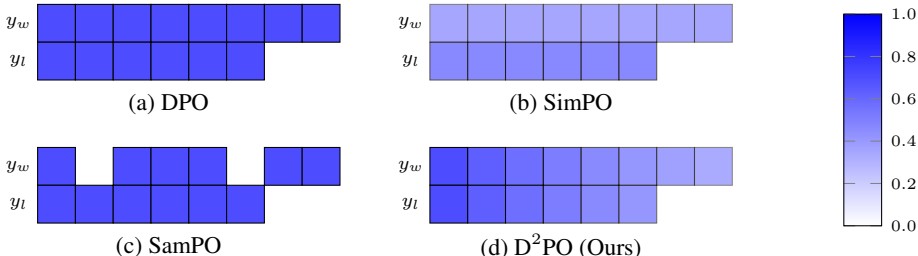

Figure 2: Illustration of coefficients in DPO, SimPO, SamPO, and our D²PO across various positions. Each box represents a coefficient, and the opacity denotes the magnitude, with darker colors indicating higher values. (a) For DPO, the coefficients are uniform across different positions. (b) For SimPO, the coefficients of the chosen $y_w$ and the rejected $y_l$ are normlaized by their lengths $|y_w|$ and $|y_l|$, respectively. (c) In SamPO, the coefficients are selected based on the minimum length of $|y_w|$ and $|y_l|$. (d) Our method introduces a $\gamma$ factor to implement coefficient decay, specifically as a sequence defined by $\gamma^t$ (e.g., 1, $\gamma$, $\gamma^2$, ..., $\gamma^T$). Here, we use $\gamma = 0.9$ for a clear visualization.

## 3 METHODOLOGY

### 3.1 DIRECT PREFERENCE OPTIMIZATION (DPO).

Direct Preference Optimization (DPO) is a pivotal advancement in the field of offline preference-based training for language models. Traditional RLHF involves a complex, multi-stage process that includes training a reward model to align with human preferences and subsequently optimizing a policy model to maximize this reward while staying close to the original model's distribution. DPO simplifies this process by reparameterizing the reward function directly in terms of the policy model, eliminating the need for an explicit reward model:

$$r(x, y) = \beta \log \frac{\pi_\theta(y|x)}{\pi_{ref}(y|x)} + \beta \log Z(x), \tag{3}$$

where $\pi_\theta$, $\pi_{ref}$ denotes the policy model and reference model, respectively. Z(x) is the partition function, and $\beta$ is a hyperparameter to control the deviation from the reference model. Substituting this reward into the Bradley-Terry (BT) ranking objective yields the DPO loss:

$$\mathcal{L}_{\text{DPO}}(\pi_\theta; \pi_{\text{ref}}) = -\mathbb{E}_{(\mathbf{x}, \mathbf{y}_w, \mathbf{y}_l) \sim \mathcal{D}} \left[ \log \sigma \left( \beta \log \frac{\pi_\theta(\mathbf{y}_w \mid \mathbf{x})}{\pi_{\text{ref}}(\mathbf{y}_w \mid \mathbf{x})} - \beta \log \frac{\pi_\theta(\mathbf{y}_l \mid \mathbf{x})}{\pi_{\text{ref}}(\mathbf{y}_l \mid \mathbf{x})} \right) \right] \tag{4}$$

DPO operates by formulating an implicit reward using the log ratio of the likelihood of a response between the current policy model and a supervised fine-tuned (SFT) model. This reward is then incorporated into the Bradley-Terry ranking objective to directly optimize the policy model for preference data. The effectiveness of DPO lies in its ability to simplify the preference optimization process, making it more accessible and efficient for practical applications.

### 3.2 TEMPORAL DECAY MATTERS IN PREFERENCE LEARNING.

**Motivation.** Preference learning plays a pivotal role in optimizing LLMs, especially when leveraging user feedback to align model outputs with human preferences. While methods like DPO (Rafailov et al., 2023) and its successors (Meng et al., 2024; Lu et al., 2024) have demonstrated significant potential in this domain, they exhibit a critical oversight: the uniform treatment of tokens across the sequence. As illustrated in Figure 2, DPO, SimPO, and SamPO assign identical coefficients to all tokens within the chosen response $y_w$ and the rejected response $y_l$. We argue that optimizing each token equally, without considering their positional importance or temporal dependence, is suboptimal for most scenarios.

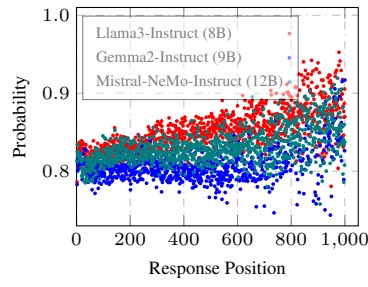

Figure 3: Probability against positions on 1000 samples.

Our observations indicate that earlier tokens receive greater optimization during the preference learning process compared to later ones (see Figure 1 ). This suggests that the benefits derived from the alignment phase over SFT are predominantly due to the optimization of initial tokens. Additionally, when plotting the prediction probability across different response positions in Figure 3, we find that more recent tokens have higher probabilities than earlier tokens. This indicates that the model becomes increasingly confident in predicting tokens as it progresses through the sequence, likely due to the accumulating contextual information from previous tokens. However, since the accuracy of these later tokens is already high—reaching up to 0.9, further improvements are more likely to come from enhancing the accuracy of the earlier tokens. Therefore, a natural approach is to focus on improving the accuracy of the prefix: *the more accurate the earlier tokens are, the better the overall quality of the sequence will be.*

**Temporal Decay Mechanism.** Inspired by the success of Yang et al. (2024), where earlier steps are crucial in the reverse chain of the diffusion denoising process, we propose a temporal decay mechanism to highlight the importance of earlier tokens in LLM scenarios. Considering the original DPO loss formulation, the most direct way to prioritize earlier tokens is to incorporate a position-dependent coefficient that decays over time. Multiple decay mechanisms can achieve this, including linear, polynomial, step, and cosine decay functions. After evaluating these options, we chose exponential decay due to its simplicity and effectiveness.

Exponential decay applies a coefficient that decreases exponentially with the token position, represented as $\gamma^t$, where $\gamma$ is the decay rate ($0 < \gamma < 1$) and $t$ is the token's position. This approach provides a smooth and gradual reduction in the influence of later tokens, ensuring that earlier tokens have a more significant impact on the loss calculation. To this end, we adapt this concept to the DPO loss function which defined as:

$$\mathcal{L}_{\mathrm{D^2PO}}\left(\pi_\theta; \pi_{\mathrm{ref}}\right) = -\log \sigma \left(\sum_{t=0}^{T_w} \gamma^t \beta \log \frac{\pi_\theta\left(\mathbf{y}_w^t \mid \mathbf{x}, \mathbf{y}_w^{<t}\right)}{\pi_{\mathrm{ref}}\left(\mathbf{y}_w^t \mid \mathbf{x}, \mathbf{y}_w^{<t}\right)} - \sum_{t=0}^{T_l} \gamma^t \beta \log \frac{\pi_\theta\left(\mathbf{y}_l^t \mid \mathbf{x}, \mathbf{y}_l^{<t}\right)}{\pi_{\mathrm{ref}}\left(\mathbf{y}_l^t \mid \mathbf{x}, \mathbf{y}_l^{<t}\right)}\right) \tag{5}$$

In this formulation, the exponential decay factor $\gamma^t$ adjusts the contribution of each token based on its position in the sequence. As is shown in Figure 2 (d), coefficients of each token in $\mathrm{D^2PO}$ gradually decrease along the position (the color from dark to light), placing greater emphasis on earlier tokens. This modification aligns the optimization process with the observed pattern of optimization in preference learning, where initial tokens benefit more from the alignment phase. Similar to the derivation of the vanilla DPO, we provide the detailed derivation of $\mathrm{D^2PO}$ in the Appendix E.

## 3.3 REFERENCE-FREE IS CONSISTENT WITH ON-POLICY SETUPS.

Reference-based methods often incorporate a KL divergence constraint to prevent the policy model from deviating significantly from its initial state, which adds computational and memory overhead. In the context of DPO, this constraint appears as an adaptive margin term $\log \frac{\pi_{\mathrm{ref}}(y_l|x)}{\pi_{\mathrm{ref}}(y_w|x)}$ in the pairwise loss function. This term quantifies the reference model's preference difference between less-preferred ($y_l$) and preferred ($y_w$) responses. We note that the DPO loss can be viewed as a specific case of contrastive loss, where $\log \pi_\theta(y)$ measures the relevance between the prompt $x$ and the response $y$. The adaptive margin ensures that loss values remain moderate, contributing to training stability. However, if the reference model assigns similar probabilities to both $y_w$ and $y_l$ (i.e., the margin approaches zero), the impact of the reference model diminishes, suggesting that it can be easily excluded.

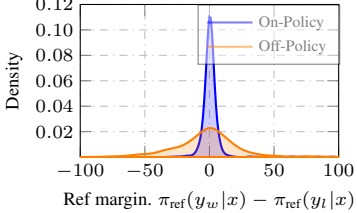

Figure 4: Reference margin of DPO.

To validate this, we analyze the margin distributions in the UltraFeedback dataset under off-policy and on-policy settings. In the off-policy setting, we use original responses, while in the on-policy setting, responses are regenerated by the same model. As illustrated in Figure 4, the on-policy dataset exhibits smaller variance in margins and an average closer to zero compared to the off-policy dataset. This indicates a higher proportion of semi-hard samples in the on-policy data. From this perspective, we can discard the KL divergence constraint under on-policy settings and easily

derive the reference-free version loss function:

$$\mathcal{L}_{\text{D}^2\text{PO}}(\pi_\theta) = -\log\sigma\left(\sum_{t=0}^{T_w}\gamma^t\beta\log\pi_\theta\left(\mathbf{y}_w^t \mid \mathbf{x}, \mathbf{y}_{\mathbf{w}}^{<\mathbf{t}}\right) - \sum_{t=0}^{T_l}\gamma^t\beta\log\pi_\theta\left(\mathbf{y}_l^t \mid \mathbf{x}, \mathbf{y}_{\mathbf{l}}^{<\mathbf{t}}\right)\right) \quad (6)$$

## 4 THEORETICAL ANALYSIS

In this section, we analyze the influence of the discount factor $\gamma$ on the performance of our method compared to DPO. Both DPO and our method can be considered as a token-level MDP that satisfies the Bellman equation. Here, we define the suboptimality as the performance difference between the optimal policy $\pi^*$ and a given policy $\pi$ under specific discount factors, which has been widely discussed in offline RL (Rashidinejad et al., 2021; Jin et al., 2021).

### 4.1 SUBOPTIMALITY DECOMPOSITION

**Definition 1 (Suboptimality).** The suboptimality of a policy $\pi$ with respect to the optimal policy $\pi^*$, starting from an initial state $s$ under discount factor $\gamma$, is defined as:

$$\text{SubOpt}(\pi, s; \gamma) = V_\gamma^{\pi^*}(s) - V_\gamma^\pi(s), \quad (7)$$

where $V_\gamma^\pi(s) = \mathbb{E}_\pi\left[\sum_{t=0}^{H-1}\gamma^t r(s_t, a_t) \mid s_0 = s\right]$ is the expected return of policy $\pi$ under discount factor $\gamma$, and $H$ is the finite horizon. To analyze the influence of the discount factor $\gamma$, we consider the suboptimality of our method evaluated with an evaluation discount factor $\gamma_e = 1.0$. We decompose the suboptimality into three terms that separately capture the differences due to the discount factors and the policy discrepancies. In this way, we can rewrite the suboptimality as below:

$$\text{SubOpt}(\pi, s; \gamma_e) = V_{\gamma_e}^{\pi^*}(s) - V_{\gamma_e}^\pi(s)$$
$$= \underbrace{\left[V_{\gamma_e}^{\pi^*}(s) - V_\gamma^{\pi^*}(s)\right]}_{\Delta_1} + \underbrace{\left[V_\gamma^{\pi^*}(s) - V_\gamma^\pi(s)\right]}_{\Delta_2} + \underbrace{\left[V_\gamma^\pi(s) - V_{\gamma_e}^\pi(s)\right]}_{\Delta_3} \quad (8)$$

This decomposition allows us to separately analyze the impact of the discount factors and the policy differences.

### 4.2 SUBOPTIMALITY ANALYSIS

**Theorem 1(Suboptimality Upper Bound).** *Let $\pi^*$ denote the optimal policy, and $\pi$ be the policy under a discount factor $\gamma \in [0, 1)$. Assume that rewards are bounded such that $|r(s, a)| \leq R$ for all states $s$ and actions $a$, and consider a finite horizon $H$. Then, the suboptimality of $\pi$ compared to $\pi^*$ when evaluated with an evaluation discount factor $\gamma_e = 1.0$ satisfies the following upper bound:*

$$\text{SubOpt}(\pi, s; \gamma_e) \leq 2\left(H - \frac{1-\gamma^H}{1-\gamma}\right)R + \frac{2(1-\gamma^H)^2}{(1-\gamma)^2}E_{s\sim d^{\pi^*}}\left[\mathbb{TV}(\pi^*(a|s)||\pi(a|s))\right]R \quad (9)$$

The complete proof is included in Appendix F. This upper bound reveals that the suboptimality depends on both the discount factor $\gamma$ and the mismatch between $\pi$ and $\pi^*$. Specifically, the first term $H - \frac{1-\gamma^H}{1-\gamma}$ decreases as $\gamma$ increases, while the second term $\left(\frac{1-\gamma^H}{1-\gamma}\right)^2$ increases, highlighting a trade-off in the choice of $\gamma$. As both terms vary monotonically with the discount factor $\gamma$ but in opposing directions, there exists an optimal value $\gamma^*$ within the interval $(0, 1)$ that balances these effects to minimize the overall suboptimality.

## 5 EXPERIMENTS

### 5.1 EXPERIMENTAL SETUPS

Due to page limitations, we briefly describe the model setting, training data, and hyperparameters in the following section. Expanded details on evaluation benchmarks and baselines are available in Appendix A.

Table 1: We report AlpacaEval 2 (Li et al., 2023) (denoted by AE2), Arena-Hard (Li et al., 2024) (denoted by AH), and MT-Bench (Zheng et al., 2023) (denoted by MB) results under three settings using standard provided samples. Note that LC and WR denote length-controlled and raw win rate, respectively. We used off-the-shelf models as the SFT model. And our judge model is GPT-4-Turbo.

| Method | Llama3-Instruct (8B) AE2 | | AH | MB | Gemma2-Instruct (9B) AE2 | | AH | MB | Mistral-NeMo-Instruct (12B) AE2 | | AH | MB |
|---|---|---|---|---|---|---|---|---|---|---|---|---|
| | WR (%) | LC (%) | WR (%) | G4-T | WR (%) | LC (%) | WR (%) | G4-T | WR (%) | LC (%) | WR (%) | G4-T |
| SFT | 39.1 | 40.1 | 27.6 | 7.5 | 37.6 | 47.2 | 44.1 | 8.3 | 44.6 | 47.7 | 46.5 | 8.1 |
| DPO | 37.4 | 40.3 | 27.7 | **7.7** | 38.8 | 48.8 | 42.5 | 8.1 | 44.4 | 49.3 | 48.5 | 8.3 |
| KTO | 33.3 | 38.1 | 21.0 | 7.5 | 39.1 | 50.0 | 43.8 | 8.3 | 37.4 | 48.7 | 35.8 | 8.2 |
| IPO | 42.2 | **45.7** | 31.9 | 7.6 | 41.0 | 50.0 | 48.2 | 8.0 | 39.8 | 48.9 | 39.8 | 8.2 |
| SamPO | 40.7 | 43.1 | 26.1 | 7.5 | 39.9 | 50.1 | 46.9 | 8.2 | 43.5 | 49.5 | 50.1 | 8.1 |
| $D^2$PO (ours) | **43.5** | 43.0 | **37.0** | 7.7 | **45.5** | **51.0** | **50.2** | **8.3** | **51.3** | **54.4** | **51.8** | **8.4** |
| ORPO | 10.6 | 15.3 | 6.8 | 6.3 | 11.3 | 21.6 | 10.2 | 7.1 | 9.6 | 17.0 | 9.8 | 6.9 |
| SimPO | 0.3* | 0.8* | 1.4* | 1.6* | 38.8 | 50.0 | 31.6 | 8.0 | 46.8 | **53.3** | 46.6 | 8.0 |

Table 2: Following the setting in Meng et al. (2024), we used the on-policy data to obtain the chosen and rejected and applied a stronger reward model. † denotes our reference-free version.

| Method | Llama3-Instruct (8B) AE2 | | AH | MB | Gemma2-Instruct (9B) AE2 | | AH | MB | Mistral-NeMo-Instruct (12B) AE2 | | AH | MB |
|---|---|---|---|---|---|---|---|---|---|---|---|---|
| | WR (%) | LC (%) | WR (%) | G4-T | WR (%) | LC (%) | WR (%) | G4-T | WR (%) | LC (%) | WR (%) | G4-T |
| SFT | 39.1 | 40.1 | 27.6 | 7.5 | 37.6 | 47.2 | 44.1 | 8.3 | 44.6 | 47.7 | 46.5 | 8.1 |
| DPO | 46.2 | 47.6 | 42.4 | 7.8 | 47.0 | 53.4 | 56.7 | **8.4** | 53.5 | 53.3 | 59.0 | 8.4 |
| KTO | 42.4 | 44.8 | 32.1 | 7.7 | 48.3 | 53.4 | 57.1 | 8.3 | 48.9 | 51.9 | 53.2 | 8.4 |
| IPO | 42.9 | 46.0 | 34.5 | 7.9 | 50.9 | 50.0 | 59.7 | 8.3 | 53.6 | 54.4 | 59.7 | 8.4 |
| SamPO | 44.4 | 47.2 | 35.8 | **8.0** | 45.8 | 55.2 | 55.2 | 8.2 | 51.1 | 53.0 | 58.3 | 8.3 |
| $D^2$PO (ours) | **47.4** | **53.5** | **47.3** | 7.8 | **57.2** | **59.7** | **66.4** | 8.3 | **57.3** | **62.1** | **62.3** | **8.6** |
| ORPO | 37.8 | 39.3 | 25.5 | 7.7 | 41.9 | 51.1 | 45.3 | 8.2 | 43.8 | 47.5 | 46.0 | 8.2 |
| SimPO | 44.4 | 50.3 | 41.9 | **7.8** | 54.5 | 58.4 | 65.0 | **8.3** | 51.3 | 55.0 | 61.9 | **8.3** |
| $D^2$PO† (ours) | **48.0** | **53.9** | **46.1** | 7.7 | **56.7** | **60.8** | **65.7** | 8.3 | **58.3** | **62.4** | **63.6** | 8.3 |

**Model Setting.** We conducted preference optimization experiments using three model families: Llama3-8B (AI@Meta, 2024), Gemma2-9B (Team et al., 2024) and Mistral-12B (Jiang et al., 2023). Here, we mainly focused on building our systems upon the instruct models. Thus, we utilized pre-trained instruction-tuned models (e.g., meta-llama/Meta-Llama-3-8B-Instruct, google/gemma-2-9b-it, and nvidia/Mistral-NeMo-12B-Instruct) as the SFT models.[1]

**Training Data** Our experiments were carried out using the UltraFeedback dataset. Specifically, We categorize the preference data into two types: 1) off-policy data (original response pairs from the UltraFeedback dataset), and 2) on-policy data generated using the SFT models. Similar to SimPO (Meng et al., 2024), for each prompt $x$, we generated 5 responses using the SFT model with a sampling temperature of 0.8. To validate these responses, we employed RLHFlow/ArmoRM-Llama3-8B-v0.1 (Wang et al., 2024c) to assign scores to each response, allowing us to select the highest-scoring response as $y_w$ and the lowest-scoring one as $y_l$.

**Hyperparameters** For all models, we set the maximum response length to 2,048 tokens and used a batch size of 128. Optimization was performed using the AdamW optimizer (Kingma & Ba, 2015) with a learning rate of $5e-7$ and a cosine learning rate schedule featuring a 10% warmup period. In preference optimization methods, including DPO and its variants such as our method $D^2$PO and SamPO, we set $\beta$ to 0.1 to ensure a fair comparison.

---

[1]The exact nature of the instruction-tuning (whether it includes SFT or the complete RLHF pipeline) of these models is not fully disclosed. For simplicity, we refer to these as SFT models.

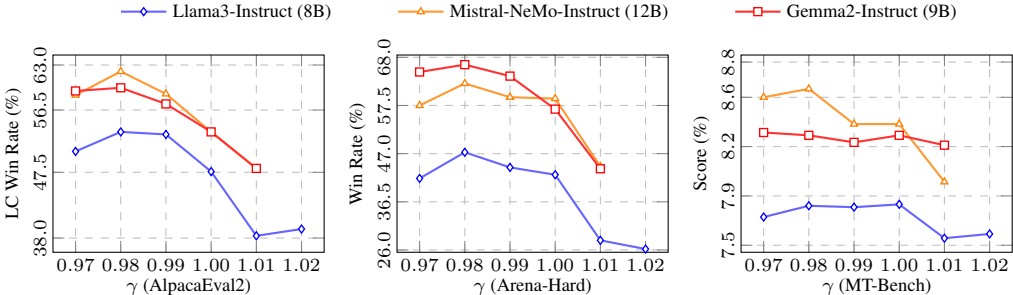

Figure 5: Performance against different $\gamma$ choices of three open-source models on three benchmarks.

## 5.2 EXPERIMENTAL RESULTS

In our experiments, we provide a comprehensive comparison of our proposed method against DPO and its variants on both off-policy and on-policy data respectively, along with the baselines introduced in Section 5.1. The baselines are categorized into two broad paradigms: reference-based and reference-free (SimPO and ORPO). Notably, as discussed in Section 3.3, our method can be seamlessly integrated into the reference-free paradigm using on-policy data. We ensure fair comparisons by maintaining consistency in the codebase and experimental settings across all methods evaluated.

**Off-policy Setups.** Table 1 clearly demonstrates that our method delivers significant improvements in win rates across all configurations. Specifically, when applied to the Llama3, our method outperforms DPO by margins of 6.1% and 2.9% in standard and length-controlled evaluation scenarios, respectively. Similarly, for the Mistral-NeMo model, our method surpasses DPO by margins of 6.9% and 5.1% in standard and length-controlled scenarios, respectively. We observed that reference-free methods exhibited instability when applied to off-policy data, often leading to a degradation in model performance. This issue is particularly evident with SimPO, where previous work observed similar findings (Lu et al., 2024). This phenomenon highlights the challenges associated with reference-free methods in preference optimization on off-policy data.

**On-policy Setups.** As shown in Table 2, our proposed method, along with all baselines, achieves better results compared to off-policy settings. Notably, our method consistently demonstrates improvements across different setups. Due to the reward model's length preference when selecting on-policy data, models trained on this data are more prone to verbosity. A critical observation in standard evaluations is the inherent bias favoring models that generate longer responses, which tend to achieve higher win rates. However, our method not only achieves superior win rates but also produces significantly shorter responses, showcasing its efficiency in generating concise and relevant outputs. Additionally, when the reference model is omitted, our method outperforms SimPO by 2.4–7.4 in LC win rate and 0.7–4.2 in win rate on AlpacaEval 2 and Arena-Hard, respectively. These findings further underscore the robustness and effectiveness of our approach. This superiority in both reference-based and reference-free contexts emphasizes the versatility and reliability of our method in preference optimization.

## 6 ANALYSES

$\gamma$ **Plays an Important Role.** The temporal decay is one of the main contributions of this work, and we would like to show how the $\gamma$ affects the performance. We conducted ablation studies on three open-source models for robust conclusions. Through results as shown in Figure 5, we see that nearly all variants with $\gamma$ lower than 1.0 consistently outperform DPO[2]. Also, $\gamma = 0.98$ achieves the highest performance across three benchmarks for these strong open-source models. This indicates that our method is robust to the choice of $\gamma$, reducing the need for extensive hyperparameter tuning.

$\gamma$ **Larger than 1 is Harmful.** As highlighted in the previous section, we prioritize earlier feedback over more recent feedback, aligning with the next-token prediction paradigm. We conducted an experiment where the decay factor $\gamma$ was set to slightly greater than 1.0 to observe the effects. The

---

[2]DPO is a special case of ours where $\gamma$ equals to 1.0.

Table 3: Results on OpenLLM Benchmark, including reasoning and mathematical testsets. Note that Hella. denotes Hellaswag, Truth. denotes TruthfulQA and Wino. denotes Winogrande.

| Method | MMLU | GSM8K | Math | IFEval | ARC-C | Hella. | Truth. | Wino. |
|---|---|---|---|---|---|---|---|---|
| | 0-shot | 0-shot | 0-shot | 0-shot | 25-shot | 10-shot | 0-shot | 5-shot |
| **(a) Llama3-Instruct (8B)** | | | | | | | | |
| SFT | 61.7 | 78.5 | 7.9 | 68.6 | 62.0 | 78.8 | 51.6 | 75.5 |
| DPO | 56.7 | 70.5 | 7.8 | 65.1 | 65.1 | **79.9** | 56.4 | 74.5 |
| SimPO | 55.2 | 57.5 | 5.3 | 60.8 | **67.6** | 78.8 | **63.8** | 74.3 |
| D$^2$PO (ours) | **61.4** | **72.0** | **8.5** | **65.6** | 65.8 | 79.0 | 57.6 | **75.1** |
| **(b) Gemma2-Instruct (9B)** | | | | | | | | |
| SFT | 72.8 | 87.4 | 19.4 | 71.9 | 71.8 | 81.7 | 60.2 | 77.9 |
| DPO | 72.2 | 88.5 | 19.4 | 60.1 | 69.9 | 71.5 | 57.7 | 72.7 |
| SimPO | 72.4 | 88.2 | 19.0 | **71.5** | 68.3 | 66.5 | 58.9 | 73.7 |
| D$^2$PO (ours) | **72.7** | **88.9** | **21.2** | 71.2 | **71.4** | **81.0** | 61.3 | **76.0** |

results could also be observed in Figure 5. When $\gamma$ exceeds 1.0, rewards linked to later tokens in the sequence receive larger coefficients than those for earlier tokens. However, this adjustment was detrimental to preference optimization, resulting in performance that lagged behind the standard DPO on both the AlpacaEval 2 and Arena-Hard benchmarks. This finding demonstrates the crucial role of earlier tokens in the alignment process and indicates that overemphasizing later tokens can degrade model performance.

**Evaluations on OpenLLM Benchmark.** To verify whether the improvements of D$^2$PO on the aforementioned RLHF benchmarks, such as Alpaca Eval2, Arena Hard, and MT-bench, come at the expense of general language modeling ability, we conducted a comprehensive evaluation of downstream tasks on the Open LLM leaderboard[3]. Specifically, we employed zero-shot evaluations on MMLU (Hendrycks et al., 2021), GSM8K (Cobbe et al., 2021), MATH (Hendrycks et al., 2021), IFEval (Zhou et al., 2023), and TruthfulQA (Lin et al., 2022). Additionally, we performed few-shot evaluations on ARC-C (Clark et al., 2018), Hellaswag (Zellers et al., 2019), and Winogrande (Levesque et al., 2012) according to the official settings in the Open LLM leaderboard. The results are summarized in Table 3, and we observe that:

- In the Llama3-8B configuration, our D$^2$PO method significantly outperforms both DPO and SimPO, particularly on the MMLU and MATH benchmarks. Notably, D$^2$PO exhibits less performance degradation on GSM8K compared to SimPO, despite both methods effectively controlling output length. D$^2$PO achieves substantial performance gains on the MATH dataset, surpassing the Instruct model by 0.55 points, while the other two methods show a noticeable decline.

- In the Gemma2-9B configuration, we observe a similar pattern, with D$^2$PO demonstrating a significant performance advantage on the MATH benchmark. These results suggest that D$^2$PO effectively enhances reasoning and mathematical problem-solving abilities in LLMs across different models. Furthermore, these additional evaluations on specialized datasets confirm that D$^2$PO maintains its effectiveness across various contexts and task types.

**Comparisons of Various Decay Strategies** We have proven the importance of temporal decay. Following the classic Markov Decision Process, we use exponential decay as our default decay strategy. Meanwhile, We also consider several variants of decay strategies, including Head decay, Linear decay and Power-Law decay. The detailed decay mechanism are summarized in Table 4. We observe 1-0 decay and Linear Deacy show inferior results to the tenporal decay, and even underperforms with the vanilla DPO. While though Power-Law method also shows promising results, but it cannot properly control the response length competitive results with exponential decay.

**Lengthy Debias.** Previous studies (Park et al., 2024; Lu et al., 2024; Meng et al., 2024) have demonstrated that DPO is susceptible to length exploitation, as it tends to amplify verbosity biases present in the preference datasets. This propensity can lead to suboptimal outcomes where the

---

[3]Open LLM leaderboard is created by huggingface to provide a standardized evaluation setup for LLMs, which includes several popular benchmarks encompassing a wide range of capabilities across multiple domains.

Table 4: Comparison of different decay mechanisms in terms of performance and response length.

| Decay Strategy | Rewards | AE2 | | | AH | | MB |
|---|---|---|---|---|---|---|---|
| | | WR (%) | LC (%) | Len. | WR (%) | Len. | G4-T |
| Exponential | $\sum\limits_{t=0}^{T} \gamma^t \beta \log \frac{p_\theta(\mathbf{y}_t\|\mathbf{x},\mathbf{y}_{<t})}{p_{ref}(\mathbf{y}_t\|\mathbf{x},\mathbf{y}_{<t})}$ | 57.2 | 59.7 | 1950 | 66.4 | 724 | 8.3 |
| Head | $\sum\limits_{t=0}^{\gamma T} \beta \log \frac{p_\theta(\mathbf{y}_t\|\mathbf{x},\mathbf{y}_{<t})}{p_{ref}(\mathbf{y}_t\|\mathbf{x},\mathbf{y}_{<t})}$ | 48.6 | 54.7 | 1762 | 57.4 | 680 | 8.2 |
| Linear | $\sum\limits_{t=0}^{\gamma T} \left(1 - \frac{t}{\gamma T}\right) \beta \log \frac{p_\theta(\mathbf{y}_t\|\mathbf{x},\mathbf{y}_{<t})}{p_{ref}(\mathbf{y}_t\|\mathbf{x},\mathbf{y}_{<t})}$ | 48.3 | 54.5 | 1713 | 59.4 | 661 | 8.3 |
| Power-Law | $\sum\limits_{t=0}^{T} \frac{1}{t^\gamma} \beta \log \frac{p_\theta(\mathbf{y}_t\|\mathbf{x},\mathbf{y}_{<t})}{p_{ref}(\mathbf{y}_t\|\mathbf{x},\mathbf{y}_{<t})}$ | 56.8 | 57.7 | 2011 | 71.2 | 823 | 8.5 |

model's decisions are disproportionately influenced by the length of the responses rather than their quality or relevance. To investigate the relationship between the length bias of training data and the output length of the model, we visualized the DPO and D$^2$PO loss of 1000 random samples based on the length gap between the chosen and rejected responses. For simplicity, verbosity-biased data refers to pairs in which the chosen response must be longer than the reject response and brevity-biased data refers to the opposite type of data.

From Figure 6, we can see that during the DPO training process, the loss of verbosity-biased data is large, while the loss of brevity-biased data is small. Consequently, DPO prioritizes the optimization of verbosity-biased data, increasing likelihood of longer chosen responses and decreasing likelihood of shorter ones. This kind of imbalance loss can easily cause model verbosity. Meanwhile, D$^2$PO reduce the loss imbalance between verbosity-biased data and brevity-biased data, thereby controlling the output length of the model.

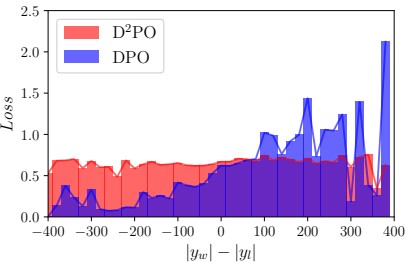

Figure 6: Loss vs. length diff.

**Human Evaluations** To further validate our results, we conducted human evaluations on the AlpacaEval2 and Arena-Hard datasets using the Gemma2-9B model. We enlisted four evaluators, with each person evaluating 50 samples for each benchmark. For each instruction, we randomized the order of the outputs from DPO and D$^2$PO to prevent bias. The evaluators assessed the responses based on three criteria: accuracy, completeness, and relevance, determining which response was better for each sample. If both responses were equally correct or incorrect, the result was considered a tie. As shown in Table 5, our comparison between D$^2$PO and DPO indicates that D$^2$PO achieved a significantly higher win rate than DPO, with an overall win rate of 67% in Arena-Hard and 69% in AlpacaEval 2 (calculated as (win + tie/2) / total).

Table 5: Human evaluation results on two benchmarks.

| Benchmark | Win | Tie | Lose |
|---|---|---|---|
| AlpacaEval 2 | 116 | 36 | 48 |
| Arena-Hard | 107 | 62 | 31 |

## 7 CONCLUSIONS

In this work, we revisited the loss objectives of DPO and its variants, introducing a temporal decay mechanism governed by a parameter $\gamma$. Motivated by the observation that earlier tokens contribute more significantly during preference optimization, our dynamic weighting scheme prioritizes these initial tokens, aligning naturally with the next-token prediction paradigm. Extensive experiments demonstrate that our approach consistently outperforms vanilla DPO, achieving notable improvements across diverse benchmarks and model architectures. By enabling DPO to focus more on short-term rewards while retaining its simplicity and stability, our method offers a compelling solution for preference-based fine-tuning of large-scale models. Furthermore, we showed that our method can be extended to a reference-free, on-policy setting, outperforming existing approaches.

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

## A EXPERIMENTAL SETUPS

**Evaluation Benchmarks.** We primarily evaluated our models using three widely used open-ended instruction-following benchmarks: MT-Bench (Zheng et al., 2023), AlpacaEval 2 (Li et al., 2023)[4], and Arena-Hard v0.1 (Li et al., 2024). These benchmarks assess the models' versatile conversational capabilities across a diverse set of queries and are widely adopted by the research community. Concretely, AlpacaEval 2 comprises 805 questions from 5 datasets, while MT-Bench encompasses 8 categories with 80 questions. Arena-Hard, an enhanced version of MT-Bench[5], includes 500 rigorously defined technical problem-solving queries. For AlpacaEval 2, we used `alpaca_eval_gpt4_turbo_fn` as the annotator which has a higher human agreement and report both the raw win rate (WR) and the length-controlled win rate (LC) (Dubois et al., 2024), with the LC metric designed to be robust against model verbosity. For Arena-Hard, we reported the WR against the baseline model. For MT-Bench, we report the average MT-Bench score, using `GPT-4-Turbo-2024-04-09` as the judge model[6].

**Baselines.** We selected several advanced preference optimization baselines, including: IPO (Azar et al., 2024) is a theoretically grounded method that avoids DPO's assumption that pairwise preferences can be replaced with pointwise rewards. KTO (Ethayarajh et al., 2024) learns from non-paired preference data. ORPO (Hong et al., 2024) introduces a reference-model-free odds ratio term to directly contrast winning and losing responses with the policy model, jointly training with the SFT objective. SimPO (Meng et al., 2024) and SamPO (Lu et al., 2024) are both designed to address the issue of model verbosity by applying length normalization. We meticulously tune the hyperparameters for each baseline and report the best performance.

**AlpacaEval 2 Annotator Choice** AlpacaEval 2 provides various evaluation templates and in the official readme recommends using `weighted_alpaca_eval_gpt4_turbo` as well as `alpaca_eval_gpt4_turbo_fn`. The former is the default annotator in AlpacaEval 2 with a human agreement rate of 65.7% and much cheaper price. In all of our evaluations, we used the latter as the annotator which has a higher agreement rate of 68.1% with human annotation data.

## B EXPERIMENTAL DETAILS

Considering DPO is a special case of D$^2$PO when $\gamma = 1.0$, to compare the effects of different decay coefficients, we conduct experiments with $\beta$ fixed at 0.1. In addition, we follow the optimal hyperparameters claimed in SimPO and our code is built on LlamaFactory (Zheng et al., 2024). Across all DAAs run, the models were trained on 32 A100 with a global batch size of 128 (4 gradient accumulation steps). The hyperparameter search range of all methods are displayed in the Table 6.

## C MORE ANALYSES

**When to Decay** In our default setting, we apply decay from the beginning of the prompt rather than from the first generated tokens. Here, we investigate the implications of these two approaches. Theoretically, during the loss computation, both the chosen token $y_w$ and the rejected token $y_l$ share the same prompt prefix, which results in distinct initial scaling coefficients for the reward at the first generated position. For illustration, if the prompt length is $l$, then in our default setting, the reward for the first generated token is scaled by $y_l$ while in the alternate setting, the scaling factor would be 1. Through results in Figure 7, we can see that both two settings achieve better results than DPO, and our default setting is much better than the other one. This indicates that proper scaling factor is very important during preference optimization.

**Effect of Different Judge Models** Here, we mainly evaluated our generated results via `GPT-4-Turbo-0409`, while previous work mainly used `GPT-4-preview-1106` instead. Results in Table 7 show that D$^2$PO delivers consistent performance gains in both two judge models.

---

[4] https://tatsu-lab.github.io/alpaca_eval/

[5] Adler et al. (2024) discussed the existence of incorrect reference answers in MT-Bench, therefore a corrected version of MT-Bench was used.

[6] `GPT-4-Turbo-2024-04-09` provides more accurate reference answers and judgments compared to GPT-4.

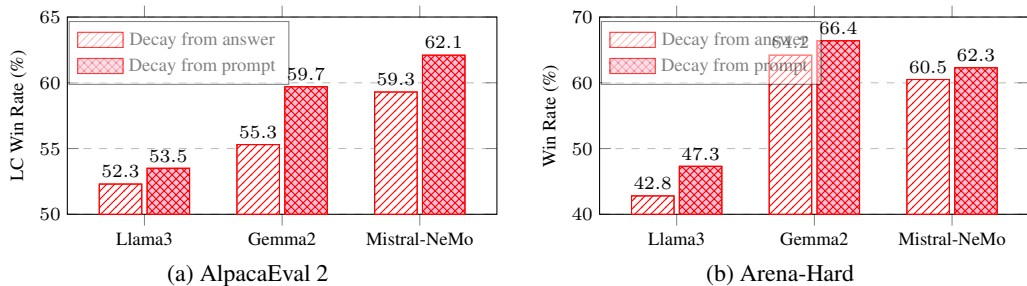

Figure 7: Visualization of the LC Win Rate (%) on three models under different decay mechanisms (answer-based and prompt-based decay) in the AlpacaEval 2 and Arena-Hard benchmarks.

Table 6: Various preference optimization objectives and hyperparameter search range.

| Method | Objective | Hyperparameter |
|---|---|---|
| DPO (Rafailov et al., 2023) | $-\log \sigma \left( \beta \log \frac{\pi_\theta(y_w|x)}{\pi_{\text{ref}}(y_w|x)} - \beta \log \frac{\pi_\theta(y_l|x)}{\pi_{\text{ref}}(y_l|x)} \right)$ | $\beta \in [0.01, 0.1]$ |
| IPO (Azar et al., 2024) | $\left( \log \frac{\pi_\theta(y_w|x)}{\pi_{\text{ref}}(y_w|x)} - \log \frac{\pi_\theta(y_l|x)}{\pi_{\text{ref}}(y_l|x)} - \frac{1}{2\tau} \right)^2$ | $\tau \in [0.01, 0.1, 0.5, 1.0]$ |
| KTO (Ethayarajh et al., 2024) | $-\lambda_w \sigma \left( \beta \log \frac{\pi_\theta(y_w|x)}{\pi_{\text{ref}}(y_w|x)} - z_{\text{ref}} \right) + \lambda_l \sigma \left( z_{\text{ref}} - \beta \log \frac{\pi_\theta(y_l|x)}{\pi_{\text{ref}}(y_l|x)} \right)$, where $z_{\text{ref}} = \mathbb{E}_{(x,y)\sim\mathcal{D}} \left[ \beta \text{KL} \left( \pi_\theta(y|x) || \pi_{\text{ref}}(y|x) \right) \right]$ | $\lambda_l = \lambda_w = 1.0$ $\beta \in [0.01, 0.05, 0.1]$ |
| SamPO (Lu et al., 2024) | $-\log \sigma \left( \sum_{t=0}^{T_m} \beta \log \frac{\pi_\theta(\mathbf{y}_w^t|\mathbf{x},\mathbf{y}_w^{<t})}{\pi_{\text{ref}}(\mathbf{y}_w^t|\mathbf{x},\mathbf{y}_w^{<t})} - \sum_{t=0}^{T_m} \beta \log \frac{\pi_\theta(\mathbf{y}_l^t|\mathbf{x},\mathbf{y}_l^{<t})}{\pi_{\text{ref}}(\mathbf{y}_l^t|\mathbf{x},\mathbf{y}_l^{<t})} \right)$ where $T_m = min(T_w, T_l), y^t \sim Uniform(T_m, y^T)$ | $\beta \in [0.01, 0.1]$ |
| ORPO (Hong et al., 2024) | $-\log p_\theta(y_w|x) - \lambda \log \sigma \left( \log \frac{p_\theta(y_w|x)}{1-p_\theta(y_w|x)} - \log \frac{p_\theta(y_l|x)}{1-p_\theta(y_l|x)} \right)$, where $p_\theta(y|x) = \exp\left( \frac{1}{|y|} \log \pi_\theta(y|x) \right)$ | $\lambda \in [0.1, 0.5, 1.0, 2.0]$ |
| SimPO (Meng et al., 2024) | $-\log \sigma \left( \frac{\beta}{|y_w|} \log \pi_\theta(y_w|x) - \frac{\beta}{|y_l|} \log \pi_\theta(y_l|x) - \gamma \right)$ | $\beta \in [2.5, 10]$ $\gamma \in [0.3, 0.5, 1.0]$ |
| D²PO | $-\log \sigma \left( \sum_{t=0}^{T_w} \gamma^t \beta \log \frac{\pi_\theta(\mathbf{y}_w^t|\mathbf{x},\mathbf{y}_w^{<t})}{\pi_{\text{ref}}(\mathbf{y}_w^t|\mathbf{x},\mathbf{y}_w^{<t})} - \sum_{t=0}^{T_l} \gamma^t \beta \log \frac{\pi_\theta(\mathbf{y}_l^t|\mathbf{x},\mathbf{y}_l^{<t})}{\pi_{\text{ref}}(\mathbf{y}_l^t|\mathbf{x},\mathbf{y}_l^{<t})} \right)$ | $\beta \in [0.1]$ $\gamma \in [0.95, 0.97, 0.98, 0.99]$ |

Table 7: Three benchmarks results with on-policy setups, using gpt-4-1106-preview as the judge model. † denotes our reference-free version.

| Method | Llama3-Instruct (8B) | | | Gemma2-Instruct (9B) | | | Mistral-NeMo-Instruct (12B) | | |
|---|---|---|---|---|---|---|---|---|---|
| | AE2 | | AH | AE2 | | AH | AE2 | | AH |
| | WR (%) | LC (%) | WR (%) | WR (%) | LC (%) | WR (%) | WR (%) | LC (%) | WR (%) |
| SFT | 31.6 | 31.7 | 19.7 | 37.7 | 48.2 | 39.9 | 40.8 | 44.2 | 39.7 |
| DPO | 41.7 | 42.9 | 31.2 | 46.4 | 53.1 | 47.4 | 53.4 | 52.6 | 47.4 |
| KTO | 35.7 | 37.7 | 25.2 | 46.5 | 52.3 | 49.2 | 45.9 | 49.7 | 45.4 |
| IPO | 40.1 | 43.2 | 25.2 | 49.1 | 48.3 | 49.5 | 51.7 | 52.9 | **51.8** |
| SamPO | 39.4 | 41.9 | 28.7 | 46.9 | 56.7 | 50.4 | 49.8 | 52.5 | 50.4 |
| D²PO (ours) | **44.5** | **50.1** | **34.1** | **59.3** | **62.3** | **58.4** | **55.0** | **60.6** | 50.6 |
| ORPO | 31.5 | 32.5 | 20.9 | 39.8 | 48.2 | 41.3 | 40.1 | 44.5 | 41.2 |
| SimPO | 44.5 | 49.1 | **33.1** | 55.1 | 59.4 | 56.5 | 50.7 | 53.8 | **51.0** |
| D²PO† (ours) | **45.0** | **51.9** | 33.0 | **58.0** | **61.5** | 56.9 | 56.8 | 59.9 | 49.4 |

**Full evaluation results**  We present the full evaluation of AlpacaEval 2, Arena-Hard and MT-Bench in Table 9 and Table 10. The former is an off-policy setup, while the latter is an on-policy setup. For on-policy setups, we found that DPO can achieve better performance when $beta = 0.01$ and reported this result for fair comparison. Specifically, "-" indicates that the model suffered a collapse during training.

Table 9: Full results on benchmarks under off-policy setups.

| Method | AlpacaEval 2 | | | Arena Hard | | MT-Bench |
|---|---|---|---|---|---|---|
| | Win Rate (%) | LC Win Rate (%) | Len. | Win Rate (%) | Len. | G4-Turbo |
| **Llama3-Instruct (8B)** | | | | | | |
| SFT | 39.05 | 40.13 | 1971 | 27.6 | 581 | 7.5 |
| DPO | 37.38 | 40.28 | 1880 | 27.7 | 546 | **7.7** |
| KTO | 33.29 | 38.06 | 1765 | 21.0 | 525 | 7.5 |
| IPO | 42.16 | 45.66 | 1845 | 31.9 | 542 | 7.6 |
| SamPO | 40.68 | 43.11 | 1891 | 26.1 | 550 | 7.5 |
| D$^2$PO ($\gamma$=0.95) | 45.90 | 44.78 | 2113 | 40.0 | 636 | 8.0 |
| D$^2$PO ($\gamma$=0.97) | 43.46 | 43.04 | 1994 | 37.0 | 602 | 7.7 |
| D$^2$PO ($\gamma$=0.98) | 42.73 | 44.10 | 1954 | 35.1 | 578 | 7.9 |
| D$^2$PO ($\gamma$=0.99) | 41.74 | 44.03 | 1912 | 30.4 | 560 | 8.0 |
| D$^2$PO ($\gamma$=1.01) | 38.50 | 40.21 | 1928 | 26.1 | 569 | 7.5 |
| D$^2$PO ($\gamma$=1.02) | 37.69 | 38.63 | 1955 | 26.8 | 572 | 7.4 |
| ORPO | 10.62 | 15.32 | 1386 | 6.8 | 764 | 6.3 |
| SimPO | 0.25 | 0.80 | 27 | 1.4 | 15 | 1.6 |
| **Gemma2-Instruct (9B)** | | | | | | |
| SFT | 37.58 | 47.23 | 1566 | 44.1 | 608 | 8.3 |
| DPO | 38.81 | 48.83 | 1546 | 42.5 | 595 | 8.1 |
| KTO | 39.07 | 50.00 | 1530 | 43.8 | 540 | 8.3 |
| IPO | 41.04 | 50.03 | 1630 | 48.2 | 608 | 8.1 |
| SamPO | 39.86 | 50.06 | 1574 | 46.9 | 596 | 8.2 |
| D$^2$PO ($\gamma$=0.95) | 48.07 | 50.05 | 1929 | 53.4 | 657 | 8.5 |
| D$^2$PO ($\gamma$=0.97) | 45.34 | 49.70 | 1824 | 50.7 | 636 | 8.3 |
| D$^2$PO ($\gamma$=0.98) | 45.46 | 50.99 | 1746 | 50.2 | 625 | 8.3 |
| D$^2$PO ($\gamma$=0.99) | 42.10 | 50.05 | 1636 | 50.2 | 609 | 8.4 |
| D$^2$PO ($\gamma$=1.01) | 38.57 | 47.45 | 1577 | 43.7 | 612 | 8.3 |
| D$^2$PO ($\gamma$=1.02) | - | - | - | - | - | - |
| ORPO | 11.30 | 21.55 | 1182 | 10.2 | 641 | 7.1 |
| SimPO | 38.76 | 50.00 | 1508 | 31.6 | 475 | 8.0 |
| **Mistral-NeMo-Instruct (12B)** | | | | | | |
| SFT | 44.60 | 47.71 | 1879 | 46.5 | 575 | 8.1 |
| DPO | 44.41 | 49.25 | 1821 | 48.5 | 569 | 8.3 |
| KTO | 37.39 | 48.68 | 1620 | 35.8 | 501 | 8.2 |
| IPO | 39.75 | 48.85 | 1634 | 39.8 | 506 | 8.2 |
| SamPO | 43.54 | 49.47 | 1784 | 50.1 | 562 | 8.1 |
| D$^2$PO ($\gamma$=0.95) | 52.17 | 52.42 | 2017 | 54.2 | 590 | 8.3 |
| D$^2$PO ($\gamma$=0.97) | 51.30 | 54.43 | 1879 | 51.8 | 562 | 8.4 |
| D$^2$PO ($\gamma$=0.98) | 49.57 | 55.43 | 1778 | 47.8 | 534 | 8.3 |
| D$^2$PO ($\gamma$=0.99) | 46.96 | 53.86 | 1770 | 45.9 | 538 | 8.0 |
| D$^2$PO ($\gamma$=1.01) | 43.65 | 47.60 | 1829 | 46.8 | 564 | 8.1 |
| D$^2$PO ($\gamma$=1.02) | 43.84 | 47.91 | 1840 | 45.6 | 572 | 8.0 |
| ORPO | 9.64 | 17.00 | 1185 | 9.8 | 640 | 6.9 |
| SimPO | 46.77 | 53.28 | 1704 | 46.6 | 500 | 8.0 |

**Arena-Hard Style Control Evaluation** We have conducted evaluations using the Arena-Hard benchmark, focusing on style control capabilities based on Gemma2-9B. Through results in Table 8, we can see that D$^2$PO consistently achieves superior performance, even when style control is a key factor, highlighting our method's effectiveness in style-controlled settings.

Table 8: Style Control evaluation on the Arena-Hard benchmark.

| Model | AH | AH (Style-Control) |
|---|---|---|
| D$^2$PO | 66.4 | 67.2 |
| DPO ($\beta$ = 0.1) | 56.7 | 57.2 |
| DPO ($\beta$ = 0.01) | 65.2 | 66.4 |
| SimPO | 65.0 | 66.3 |

Table 10: Full results on benchmarks under on-policy setups. † denotes our reference-free version.

| Method | AlpacaEval2 | | | Arena Hard | | MT-Bench |
|---|---|---|---|---|---|---|
| | Win Rate (%) | LC Win Rate (%) | Len. | Win Rate (%) | Len. | G4-Turbo |
| **Llama3-Instruct (8B)** | | | | | | |
| SFT | 39.05 | 40.13 | 1971 | 27.6 | 581 | 7.5 |
| DPO ($\beta$=0.01) | 48.26 | 49.93 | 1937 | 45.2 | 568 | 7.8 |
| KTO | 42.36 | 44.77 | 1901 | 32.1 | 545 | 7.7 |
| IPO | 42.92 | 45.99 | 1889 | 34.5 | 553 | 7.9 |
| SamPO | 44.35 | 47.17 | 1890 | 35.8 | 536 | 8.0 |
| D$^2$PO ($\gamma$=0.95) | 48.13 | 51.53 | 1832 | 42.5 | 578 | 7.7 |
| D$^2$PO ($\gamma$=0.97) | 46.15 | 50.52 | 1739 | 41.6 | 549 | 7.7 |
| D$^2$PO ($\gamma$=0.98) | 47.39 | 53.54 | 1705 | 47.3 | 518 | 7.8 |
| D$^2$PO ($\gamma$=0.99) | 48.01 | 52.97 | 1739 | 44.0 | 514 | 7.8 |
| DPO ($\beta$=0.1) | 46.21 | 47.60 | 1971 | 42.4 | 627 | 7.9 |
| D$^2$PO ($\gamma$=1.01) | 37.25 | 38.32 | 1948 | 28.1 | 578 | 7.6 |
| D$^2$PO ($\gamma$=1.02) | 37.75 | 39.30 | 1942 | 26.2 | 566 | 7.6 |
| ORPO | 37.75 | 39.29 | 1934 | 25.5 | 615 | 7.7 |
| SimPO | 44.41 | 50.34 | 1704 | 41.9 | 477 | 7.8 |
| D$^2$PO$^\dagger$ ($\gamma$=0.98) | 48.01 | 53.87 | 1726 | 46.1 | 526 | 7.7 |
| **Gemma2-Instruct (9B)** | | | | | | |
| SFT | 37.58 | 47.23 | 1566 | 44.1 | 608 | 8.3 |
| DPO ($\beta$=0.01) | 54.53 | 57.05 | 1948 | 65.2 | 768 | 8.3 |
| KTO | 48.26 | 53.39 | 1775 | 57.1 | 705 | 8.3 |
| IPO | 50.86 | 50.00 | 2129 | 59.7 | 759 | 8.3 |
| SamPO | 45.78 | 55.21 | 1662 | 55.2 | 668 | 8.2 |
| D$^2$PO ($\gamma$=0.95) | 58.39 | 59.03 | 2034 | 65.5 | 739 | 8.5 |
| D$^2$PO ($\gamma$=0.97) | 56.83 | 59.25 | 1949 | 64.8 | 715 | 8.3 |
| D$^2$PO ($\gamma$=0.98) | 57.20 | 59.71 | 1950 | 66.4 | 724 | 8.3 |
| D$^2$PO ($\gamma$=0.99) | 53.98 | 57.38 | 1843 | 63.9 | 693 | 8.2 |
| DPO ($\beta$=0.1) | 47.02 | 53.43 | 1737 | 56.7 | 682 | 8.3 |
| D$^2$PO ($\gamma$=1.01) | 38.70 | 48.06 | 1592 | 43.7 | 610 | 8.2 |
| D$^2$PO ($\gamma$=1.02) | - | - | - | - | - | - |
| ORPO | 41.93 | 51.14 | 1647 | 45.3 | 641 | 8.2 |
| SimPO | 54.47 | 58.42 | 1871 | 65.0 | 744 | 8.3 |
| D$^2$PO$^\dagger$ ($\gamma$=0.98) | 56.71 | 60.76 | 1894 | 65.7 | 687 | 8.3 |
| **Mistral-Nemo-Instruct (12B)** | | | | | | |
| SFT | 44.60 | 47.71 | 1879 | 46.5 | 575 | 8.1 |
| DPO ($\beta$=0.01) | 58.76 | 57.29 | 2160 | 63.6 | 659 | 8.3 |
| KTO | 48.26 | 53.39 | 1775 | 57.1 | 705 | 8.3 |
| IPO | 50.86 | 50.00 | 2129 | 59.7 | 759 | 8.3 |
| SamPO | 45.78 | 55.21 | 1662 | 55.2 | 668 | 8.2 |
| D$^2$PO ($\gamma$=0.95) | 59.25 | 57.85 | 2167 | 60.7 | 665 | 8.6 |
| D$^2$PO ($\gamma$=0.97) | 56.77 | 58.65 | 1969 | 57.5 | 586 | 8.6 |
| D$^2$PO ($\gamma$=0.98) | 57.34 | 62.07 | 1853 | 62.3 | 546 | 8.6 |
| D$^2$PO ($\gamma$=0.99) | 54.29 | 58.83 | 1816 | 59.3 | 532 | 8.4 |
| DPO ($\beta$=0.1) | 53.48 | 53.32 | 2081 | 59.0 | 624 | 8.4 |
| D$^2$PO ($\gamma$=1.01) | 45.34 | 48.06 | 1908 | 44.2 | 581 | 8.0 |
| D$^2$PO ($\gamma$=1.02) | - | - | - | - | - | - |
| ORPO | 41.93 | 51.14 | 1647 | 45.3 | 641 | 8.2 |
| SimPO | 54.47 | 58.42 | 1871 | 65.0 | 744 | 8.3 |
| D$^2$PO$^\dagger$ ($\gamma$=0.98) | 56.71 | 60.76 | 1894 | 65.7 | 687 | 8.3 |

**Stronger Instruct Model Evaluation**    To verify the robustness properties of our method, we conducted experiments under on-policy setups based on a stronger Instruct model, Gemma2-Instruct (27B). Considering the limited computing resources, we only compared D²PO with DPO and SimPO. Table 11 shows the evaluation results on three benchmarks, demonstrating that our method maintains a certain advantage over stronger models.

Table 11: Gemma2-Instruct (27B) results under on-policy setups

| Method | AlpacaEval2 | | | Arena Hard | | MT-Bench |
| --- | --- | --- | --- | --- | --- | --- |
| | Win Rate (%) | LC Win Rate (%) | Len. | Win Rate (%) | Len. | G4-Turbo |
| SFT | 39.68 | 48.13 | 1633 | 60.7 | 648 | 8.5 |
| DPO ($\beta$=0.1) | 48.45 | 53.66 | 1786 | 71.2 | 715 | 8.5 |
| DPO ($\beta$=0.01) | 57.39 | 59.05 | 1990 | 78.6 | 811 | **8.6** |
| SimPO | 54.29 | **61.30** | 1748 | 75.3 | 682 | 8.5 |
| D²PO ($\gamma$=0.98) | **57.52** | 60.52 | 1893 | **79.8** | 708 | **8.6** |

## D  CASE ANALYSIS

We list two concrete cases from AlpacaEval2 to compare D²PO with DPO. Regarding the first case in Figure 8, the DPO model's output overthinks and mistakenly assumes that the stopover time is already included in the total travel time in the second step of its analysis, resulting in the omission of this time in the final calculation. In contrast, the model trained with our method succinctly lists the times and correctly sums them to obtain the total travel time. Figure 9 shows the second case: given the question, "What is the largest ocean in the world?", the DPO model provides a response without explanation. In contrast, the D²PO model produces a well-structured answer with detailed explanations, making the answer more credible.

Figure 8: A case examples of AlpacaEval2, generated by Gemma2-Instruct (9B).

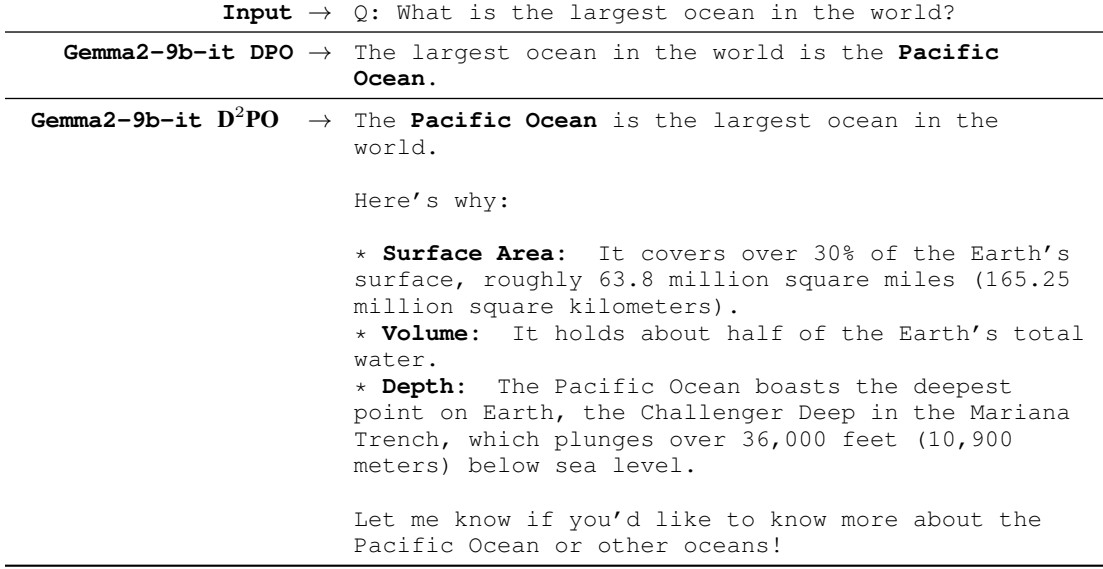

Figure 9: A case examples of AlpacaEval2, generated by Mistral-Nemo-Instruct (12B).

| | |
|---|---|
| **Input** $\rightarrow$ | Q: if it takes 8 hours to fly to Singapore, stop over for an hour and a half and then fly 14 hours to London then what is the total travel time if the plane is delayed in Singapore? |
| **Mistral-Nemo DPO** $\rightarrow$ | To calculate the total travel time including the delay in Singapore, we need to add up all the components of the journey:
1.  Flight to Singapore:  8 hours
2.  Stopover in Singapore:  1.5 hours (which is already included in the total time as it's part of the journey, not a delay)
3.  Delay in Singapore:  Let's assume the delay is 'D' hours.
4.  Flight from Singapore to London:  14 hours

The total travel time including the delay would be:

Total travel time = Flight to Singapore + Delay in Singapore + Flight to London
Total travel time = 8 hours + D hours + 14 hours
Without knowing the exact duration of the delay (D), we can express the total travel time as:
Total travel time = 22 hours + D

For example, if the delay in Singapore is 2 hours, the total travel time would be:

Total travel time = 22 hours + 2 hours = 24 hours
So, the total travel time including the delay is 22 hours plus the duration of the delay (D). |
| **Mistral-Nemo $D^2PO$** $\rightarrow$ | To calculate the total travel time including the delay in Singapore, let's break down the journey and add the delay:

Total initial travel time without delay:
– Flight to Singapore:  8 hours
– Stopover in Singapore:  1.5 hours
– Flight from Singapore to London:  14 hours

Total without delay = 8 + 1.5 + 14 = 23.5 hours

Now, let's assume the plane is delayed in Singapore by 'D' hours.  The delay will only affect the stopover time, so we add 'D' to the stopover duration:

Adjusted stopover time with delay = 1.5 hours + D hours
Now, the total travel time with the delay is:
Total with delay = 8 hours (flight to Singapore) + (1.5 + D) hours (stopover with delay) + 14 hours (flight to London)
Total with delay = 8 + (1.5 + D) + 14
Total with delay = 23.5 + D hours
Without knowing the specific duration of the delay (D), we can express the total travel time with the delay as:
Total travel time with delay = 23.5 hours + Delay in Singapore (D hours) |

# E  DERIVATION OF D²PO

In the reinforcement learning scenario, two fundamental concepts are the state-value function V and the action-value function Q. The former represents the expected cumulative reward from taking action a in state s, meanwhile the latter represents the reward under state s. We extend the relation between the Q-function and the V-function under a KL divergence constraint, as proposed in Rafailov et al. (2024), by incorporating the temporal decay mechanism:

$$Q^*\left(\mathbf{s}_t, \mathbf{a}_t\right) = \begin{cases} r\left(\mathbf{s}_t, \mathbf{a}_t\right) + \beta \log \pi_{\text{ref}}\left(\mathbf{a}_t \mid \mathbf{s}_t\right) + \gamma V^*\left(\mathbf{s}_{t+1}\right), & \text{if } \mathbf{s}_{t+1} \text{ is not terminal} \\ r\left(\mathbf{s}_t, \mathbf{a}_t\right) + \beta \log \pi_{\text{ref}}\left(\mathbf{a}_t \mid \mathbf{s}_t\right), & \text{if } \mathbf{s}_{t+1} \text{ is terminal} \end{cases} \tag{10}$$

where $\gamma \in (0, 1]$ represents the discount factor. In the assumption of DPO and its subsequent variants, $\gamma$ is typically set to 1 which indicates long-term returns do not need to decay. However, in auto-regressive scenarios such as language models, the longer the context provided, the lower the uncertainty of the model's predictions (see Figure 3). Therefore, the tokens at the beginning of the response make greater contributions to the total return. Based on this assumption, we can get the formulation of the reward over a trajectory $\tau = \{s_1, a_1, ..., a_{T-1}, s_T\}$:

$$\sum_{t=0}^{T-1} \gamma^t r(s_t, a_t) = \sum_{t=0}^{T-1} \gamma^t Q^*(s_t, a_t) - \gamma^t \beta \log \pi_{ref}(a_t, s_t) - \gamma^{t+1} V^*(s_{t+1}) \tag{11}$$

Noting that, in the general maximum entropy RL setting (Ziebart et al., 2008; Ziebart, 2010), the optimal policy is given by Boltzmann probability distribution as:

$$\pi^*\left(\mathbf{a}_t \mid \mathbf{s}_t\right) = e^{(Q^*(\mathbf{s}_t, \mathbf{a}_t) - V^*(\mathbf{s}_t))/\beta} \tag{12}$$

By taking the logarithm of the Eq. (12), we can simplified Eq. (11):

$$\sum_{t=0}^{T-1} \gamma^t r(s_t, a_t) = Q^*(s_0, a_0) - \beta \log \pi_{ref}(a_0, s_0) + \sum_{t=1}^{T-1} (\gamma^t \beta \log \frac{\pi_\theta(a_t, s_t)}{\pi_{ref}(a_t, s_t)}) \tag{13}$$

$$= V^*(s_0) + \sum_{t=1}^{T-1} (\gamma^t \beta \log \frac{\pi_\theta(a_t, s_t)}{\pi_{ref}(a_t, s_t)}) \tag{14}$$

We can then plug Eq. (14) into the Bradley-Terry ranking objective, which yields our final loss formation:

$$\mathcal{L}_{\text{D}^2\text{PO}}\left(\pi_\theta\right) = -\log \sigma \left(\sum_{t=0}^{T_w} \gamma^t r(s_t, a_t) - \sum_{t=0}^{T_l} \gamma^t r(s_t, a_t)\right) \tag{15}$$

$$= -\log \sigma \left(\sum_{t=0}^{T_w} \gamma^t \beta \log \frac{\pi_\theta\left(\mathbf{y}_w^t \mid \mathbf{x}, \mathbf{y}_w^{<t}\right)}{\pi_{\text{ref}}\left(\mathbf{y}_w^t \mid \mathbf{x}, \mathbf{y}_w^{<t}\right)} - \sum_{t=0}^{T_l} \gamma^t \beta \log \frac{\pi_\theta\left(\mathbf{y}_l^t \mid \mathbf{x}, \mathbf{y}_l^{<t}\right)}{\pi_{\text{ref}}\left(\mathbf{y}_l^t \mid \mathbf{x}, \mathbf{y}_l^{<t}\right)}\right) \tag{16}$$

This is similar to the standard DPO objective, except for an additional temporay decay term $\gamma$. We empirically set $\gamma < 1$ to focus more on short term return rather than long term return.

## F  THEORETICAL ANALYSIS

In this section, we provide the detailed derivation of the upper bound for

$$
\begin{aligned}
\text{SubOpt}(\pi, s; \gamma_e) &= V_{\gamma_e}^{\pi^*}(s) - V_{\gamma_e}^{\pi}(s) \\
&= \underbrace{\left[ V_{\gamma_e}^{\pi^*}(s) - V_{\gamma}^{\pi^*}(s) \right]}_{\Delta_1} + \underbrace{\left[ V_{\gamma}^{\pi^*}(s) - V_{\gamma}^{\pi}(s) \right]}_{\Delta_2} + \underbrace{\left[ V_{\gamma}^{\pi}(s) - V_{\gamma_e}^{\pi}(s) \right]}_{\Delta_3}
\end{aligned}
\tag{17}
$$

### F.1  THE UPPER BOUND OF $\Delta_1$ AND $\Delta 3$

Noting that $\Delta_1$ and $\Delta_3$ both capture the difference in the expected returns of the same policy when evaluated under different $\gamma$, we can analyze the upper bound of these two items together.

The term $\Delta_1$ is given by with $\gamma^e = 1.0$:

$$
\begin{aligned}
\Delta_1 &= V_{\gamma_e}^{\pi^*}(s) - V_{\gamma}^{\pi^*}(s) \\
&= \mathbb{E}_{\pi^*} \left[ \sum_{t=0}^{H-1} \gamma_e^t r(s_t, a_t) - \sum_{t=0}^{H-1} \gamma^t r(s_t, a_t) \right] \\
&= \mathbb{E}_{\pi^*} \left[ \sum_{t=0}^{H-1} (1 - \gamma^t) r(s_t, a_t) \right].
\end{aligned}
$$

Assuming the rewards are bounded, i.e., $|r(s, a)| \leq R$, we have:

$$
\Delta_1 \leq \sum_{t=0}^{H-1} (1 - \gamma^t) R = (H - \frac{1 - \gamma^H}{1 - \gamma}) R.
\tag{18}
$$

Similarly, we can obtain the upper bound of $\Delta_3$:

$$
\Delta_3 \leq \sum_{t=0}^{H-1} (1 - \gamma^t) R = (H - \frac{1 - \gamma^H}{1 - \gamma}) R.
\tag{19}
$$

### F.2  THE UPPER BOUND OF $\Delta_2$

**Lemma 1 Performance Difference Lemma with finite horizon H (Kakade & Langford, 2002)**

$$
V_{\gamma}^{\pi^*}(s) - V_{\gamma}^{\pi}(s) = \frac{1 - \gamma^H}{1 - \gamma} E_{s \sim d^{\pi^*}} \left[ \Sigma_{a \in A} (\pi^*(a \mid s) - \pi(a \mid s)) Q^{\pi}(s, a) \right]
\tag{20}
$$

where $\pi^*$ represents optimal policy and $\pi$ represents policy.

Based on the assumption that the rewards are bounded, i.e., $|r(s, a)| \leq R$, we have:

$$
Q^{\pi}(s, a) \leq \sum_{t=0}^{H-1} \gamma^t R = \frac{1 - \gamma^H}{1 - \gamma} R
\tag{21}
$$

Finally, we can get

$$
\Delta_2 = V_{\gamma}^{\pi^*}(s) - V_{\gamma}^{\pi}(s)
\tag{22}
$$

$$
\leq \frac{(1 - \gamma^H)^2}{(1 - \gamma)^2} E_{s \sim d^{\pi^*}} \left[ \sum_{a \in A} (\pi^*(a \mid s) - \pi(a \mid s)) \right] R
\tag{23}
$$

$$
= \frac{2(1 - \gamma^H)^2}{(1 - \gamma)^2} E_{s \sim d^{\pi^*}} \left[ \mathbb{TV}(\pi^*(a|s) || \pi(a|s)) \right] R
\tag{24}
$$

### F.3 SUBOPTIMAL ANALYSIS

Adding the bounds on $\Delta_1$, $\Delta_2$, and $\Delta_3$, we obtain:

$$\text{SubOpt}(\pi, s; \gamma_e) \leq 2(H - \frac{1 - \gamma^H}{1 - \gamma})R + \frac{2(1 - \gamma^H)^2}{(1 - \gamma)^2} E_{s \sim d^{\pi^*}} \left[ \mathbb{TV}(\pi^*(a|s)||\pi(a|s)) \right] R \quad (25)$$

Since both terms vary monotonically with the temporary decay factor $\gamma$ but in opposing directions, this implies the existence of an optimal trade-off value, denoted as $\gamma^*$, within the interval $(0, 1)$.

## G COMPARISON WITH STRONGLY RELATED WORK

We noticed that Yang et al. (2024) also emphasized the importance of focusing on the contribution of earlier steps, but within the reverse chain of a diffusion denoising process, rather than in an autoregressive LLM scenario. They employed a $\gamma$ parameter to control the contribution of earlier steps, ensuring the quality of generation. We would like to highlight how our approach differs from previous studies in the following perspectives:

- Different Perspectives and Tasks: While our work and the referenced prior work both involve preference optimization, they are derived from fundamentally different perspectives and are applied to different downstream tasks. The prior work focuses on text-to-image tasks, which involve fixed-length generation through a non-autoregressive diffusion process. In contrast, our research is centered on LLMs in an autoregressive context, where sequence generation dynamics are inherently different.

- Motivation Differences: As illustrated in Figures 1 and 3, our motivation diverges significantly from prior work. Our temporal decay mechanism is designed to address specific challenges in LLMs, such as length bias and the need for alignment with human preferences across varying sequence lengths.

- Flexible Decay Mechanism: Our approach is not limited to exponential decay. As shown in Table 4, we explore multiple decay strategies, demonstrating the flexibility and adaptability of our method to different scenarios and tasks.

- Theoretical Insights: We have provided a theoretical analysis based on the token-level MDP, suggesting the existence of an optimal gamma value for enhancing preference optimization. This theoretical foundation supports the practical effectiveness of our approach.

- Extension and Complementarity: Our work serves as both an extension and a complement to the referenced study. While the prior work has not validated its approach on standard RLHF benchmarks, our method has been tested and shown to be effective in these contexts, as detailed in our experimental results.

