# OpenReview forum: "Earlier Tokens Contribute More: Learning Direct Preference Optimization From Temporal Decay Perspective"
_ICLR.cc/2025/Conference — ICLR 2025 Poster_

### Official Review · Reviewer_tbQd · 2024-10-26

**Soundness:** 3
**Presentation:** 3
**Contribution:** 3
**Rating:** 6
**Confidence:** 5

**Summary:**

Most existing DPO variants assign rewards uniformly across the entire sequence, overlooking temporal dynamics in text generation.
For this purpose, this paper proposes D$^2$PO, an enhanced variant of DPO that incorporates a temporal decay factor, so as to prioritize earlier tokens that can be more critical for RLHF and focus on most relevant feedback.
Experiment results verified the gain of the proposed method over vanilla DPO.

**Strengths:**

1. The overall direction of token-level reward/credit assignment for RLHF in LMs is promising.
2. Experimental results and ablation studies are comprehensive and generally strong.

**Weaknesses:**

1. A major concern on this paper is about its contribution/novelty, as it omits several highly relevant related works. For example, the training loss eq. (5) in this paper is (almost) identical to eqs. (8) and (21) in [1]. Furthermore, the idea that earlier tokens can be more important for LMs' generation/alignment/reward optimization (e.g., L21-22, L69-71) has been discussed in [2] (first paragraph of section 3 and appendix F.3). At the minimum level, the authors ought to have a adequate citation and discussion with these prior works. Otherwise, the contribution of this paper is unjustified.

[1] Yang, Shentao, Tianqi Chen, and Mingyuan Zhou. "A Dense Reward View on Aligning Text-to-Image Diffusion with Preference." Forty-first International Conference on Machine Learning.

[2] Yang, Shentao, et al. "Preference-grounded token-level guidance for language model fine-tuning." Advances in Neural Information Processing Systems 36 (2023).

2. Missing evaluation on the OpenLLM benchmark. Does D$^2$PO's improved RLHF come at the cost of general LM ability such as MMLU and/or GSM8K?

3. L253-254: missing citation for maximum entropy RL, such as [3, 4].

[3] Ziebart, Brian D., et al. "Maximum entropy inverse reinforcement learning." AAAI. Vol. 8. 2008.

[4] Ziebart, Brian D. Modeling purposeful adaptive behavior with the principle of maximum causal entropy. Carnegie Mellon University, 2010.

**Questions:**

1. L78-79: why is the more recent feedback potentially outdated?
2. In the derivation of D$^2$PO, how did you cancel the partition function? Note that this is not an issue with the original DPO because it assume a bandit reward so that the partition function depends only on the prompt, which is the same for both chosen and rejected responses.
3. L211-212: what is the definition of "accuracy"?
4. In eq. (11), should $p_\theta$ be $\pi_\theta$?

---

> ### Author Response · Authors · 2024-11-18
> **Response to the Weakness 1**
>
> > W1: A major concern on this paper is about its contribution/novelty, as it omits several highly relevant related works.
>
> Thank you for bringing these relevant works to our attention. We acknowledge that we had not considered these papers in our original submission, and we appreciate the opportunity to discuss them and clarify our contributions in relation to prior work. After thoroughly reviewing [1] and [2], we have identified both similarities and distinctions between their approaches and ours, which we elaborate below:
>
> - Compared with [1]: While both our work and [1] refine the DPO loss by incorporating a temporal decay factor, the contexts and applications differ significantly. Our research focuses on LLMs and their alignment was mainly conducted in text-to-image generation.
>     - We emphasize optimizing earlier tokens to improve response quality and reduce length bias inherent in text generation. In contrast, [1] is centered on text-to-image (T2I) diffusion models, emphasizing the initial steps of the reverse diffusion chain in image generation.
>     - In LLMs, temporal decay across token positions can mitigate length bias because generated texts can vary in length, and users may naturally prefer more detailed replies. By introducing temporal decay, we aim to balance this tendency and enhance performance on RLHF benchmarks. In the case of diffusion models discussed in [1], the output is of a fixed size (e.g., a 64×64 image in Stable Diffusion), and both preferred and non-preferred outputs share the same dimensions. Therefore, the role and impact of temporal decay in our work differ from those in [1].
>     - Additionally, we have not seen any other work using temporal decay strategy in LLM scenario. On the other hand, ours work is a good extension of [1]. Moreover, the $\gamma$ decay is just one-of the decay mechanisms where we have already compared various decay stratiges in Table 4, including Exponential, Head, Linear and Power-Law. Note that the Power-Law decay strategy also delivers strong performance than the DPO and its variants, but have a weak constraint in length control. In a nutshell, we want to draw your attnetion that our temporal decay strategy is a learning framework, not merely a gamma-decay function.
>
> - As noted in [2], RL-based language models can be affected by token-level KL divergence and delayed feedback from sparse rewards, which may hinder optimization, especially for earlier tokens. Our method approaches this challenge differently.
>     - By utilizing the DPO framework, we avoid classical reinforcement learning altogether, eliminating issues related to delayed feedback. We directly optimize the preference between pairs of outputs, allowing for more immediate and effective adjustments to the model.
>     - Drawing inspiration from [3], we analyzed the distributional changes brought about by alignment and found that alignment significantly alters the earlier tokens, with minimal changes to the distribution of later tokens (Figure 1). In Section 3.2, we further clarify our motivation by observing that more recent tokens have higher probabilities than earlier ones. This insight led us to focus on optimizing earlier tokens, reinforcing this trend through the introduction of a temporal decay mechanism. Our empirical results demonstrate that this approach enhances LLM performance in aligning with human preferences.
>     - Another point we want to argue is that the authors only discuss the delayed-feedback issue (which is not the same with us) in the Related work section and the Appendix. It is not easy to cover all papers in this kind of large research domains as there are so many papers published years.
>
> Thank you again for your valuable feedback. Your insights have helped us improve our paper by situating our work within the existing body of research and ensuring that our comparisons are comprehensive.
>
> [1] Yang, Shentao, Tianqi Chen, and Mingyuan Zhou. "A Dense Reward View on Aligning Text-to-Image Diffusion with Preference." Forty-first International Conference on Machine Learning.
>
> [2] Yang, Shentao, et al. "Preference-grounded token-level guidance for language model fine-tuning." Advances in Neural Information Processing Systems 36 (2023).
>
> [3]THE UNLOCKING SPELL ON BASE LLMS: RETHINKING ALIGNMENT VIA IN-CONTEXT LEARNING

---

> ### Author Response · Authors · 2024-11-18
> **Response to Weakness 2&3**
>
> > W2: Missing evaluation on the OpenLLM benchmark. Does D2PO's improved RLHF come at the cost of general LM ability such as MMLU and/or GSM8K?
>
> Thank you for your thoughtful feedback and for acknowledging the effectiveness of D²PO on RLHF benchmarks. We want to assure you that the improvements brought by D²PO do not come at the expense of general language modeling abilities.
> To address your concern, we conducted additional experiments on several general language modeling benchmarks, including MMLU, GSM8K, Math, and other OpenLLM Benchmark (which includes IFEval, Hellaswag, Winogrande, TruthfulQA, and ARC-C). The results are detailed in our general response and summarized below.
>
> | Method (Llama3-8B)    | MMLU  | GSM8K | Math  | IFEval prompt-strict | ARC-C (25) | Hellaswag(10) | TruthfulQA(0)| Winogrande(5)|
> |-----------------------|-------|-------|-------|----------------------|------------|---------------|--------------|--------------|
> | Instruct              | 61.66 | 78.47 | 7.91  | 68.58                | 61.95      | 78.80         | 51.62        | 75.53        |
> | D²PO (gamma0.98)      | 61.38 | 71.95 | 8.46  | 65.62                | 65.78      | 79.03         | 57.57        | 75.14        |
> | DPO                   | 56.66 | 70.51 | 7.77  | 65.06                | 65.10      | 79.99         | 56.38        | 74.51        |
> | SimPO                 | 55.22 | 57.54 | 5.27  | 60.81                | 67.58      | 78.82         | 63.83        | 74.27        |
>
>
> | Method (Gemma2-9B)    | MMLU  | GSM8K | Math  | IFEval prompt-strict | ARC-C (25) | Hellaswag(10) | TruthfulQA(0) | Winogrande(5)|
> |-----------------------|-------|-------|-------|----------------------|------------|---------------|---------------|--------------|
> | instruct              | 72.82 | 87.41 | 19.42 | 71.90                | 71.84      | 81.67         | 60.23         | 77.90        |
> | D²PO    (gamma0.98)   | 72.68 | 88.86 | 21.22 | 71.16                | 71.42      | 81.03         | 61.34         | 76.01        |
> | DPO                   | 72.21 | 88.53 | 19.42 | 60.07                | 69.88      | 71.48         | 57.65         | 72.69        |
> | SimPO                 | 72.35 | 88.17 | 19.00 | 71.53                | 68.34      | 66.51         | 58.87         | 73.72        |
>
> Our results show that on both the Llama3-8B and Gemma2-9B models, our D²PO achieves consistent improvements on the MMLU, GSM8K, and Math benchmarks compared to the standard DPO and SimPO methods. This demonstrates that the improvement does not come at the cost of general language modeling ability, as D²PO nearly maintains performance relative to the SFT baseline (with only a slight decline in GSM8K on Llama3-8B but stronger performance on Gemma2-9B).
> Furthermore, D²PO yields surprisingly strong results on the Math benchmark, benefiting from the use of the temporal decay strategy. This strategy enables the model to generate a more precise prefix, which is crucial for arriving at the correct answer in mathematical problem-solving.
>
> Additionally, on the OpenLLM benchmarks, which are popular for evaluating pretrained LLM performance, D²PO continues to perform strongly. Notably, for Gemma2-9B, it nearly attains the same results as the SFT baseline, while the other two methods (DPO and SimPO) show noticeable performance declines. This further substantiates that D²PO enhances alignment without compromising general model capabilities.
>
>
> > W3: Missing missing citation for maximum entropy RL, such as [3, 4].
>
> Thanks for providing the two related papers for maxinum entropy RL. We will update our manuscript to include this two papers.

---

> ### Author Response · Authors · 2024-11-18
> **Response to Questions**
>
> > Q1: L78-79: why is the more recent feedback potentially outdated?
>
> We apologize for the confusion caused by our wording. We intended to emphasize that the model should prioritize earlier feedback over more recent tokens. We will revise this sentence in the manuscript.
>
> > Q2: In the derivation of D$^2$PO, how did you cancel the partition function? Note that this is not an issue with the original DPO because it assumes a bandit reward so that the partition function depends only on the prompt, which is the same for both chosen and rejected responses.
>
> The original DPO is derived as a bandit problem, while our method is considered as token-level MDP which satisfies the Bellman equation. Drawing inspirations in Rafailov et al.(2024)[1]'s work, our derivation of D²PO starts from the relationship between Q function and reward which have been explored in preference-based RL. By accumulating the temporal decay reward of each token, we get the Eq.10, and $V^*(s_0)$ can be canceled in which $y_w$ and $y_l$ start at the same state $s_0$.
>
> [1] From r to Q∗: Your Language Model is Secretly a Q-Function.
>
> > L211-212: what is the definition of "accuracy"?
>
> The accuracy represents the prediction probability of the ground truth token.
>
> > In eq. (11), should $p_\theta$ be $\pi_\theta$?
>
> Thank you for pointing out the errors in the formula, and we will fix them in the manuscript.

---

> > ### Comment · Reviewer_tbQd · 2024-11-24
> > **Reponse to the author**
> >
> > Dear authors,
> >
> > First of all, thank you for your responses and paper updates! Still, some points need to be fixed or clarified.
> >
> > ### I. About the novelty
> >
> > > 1. "Our research focuses on LLMs and their alignment was mainly conducted in text-to-image generation." "It is not easy to cover all papers in this kind of large research domains as there are so many papers published years."
> >
> > These are not a strong-enough arguments given the similarity of the proposed loss with the listed prior work(s).
> >
> > > 2. Therefore, the role and impact of temporal decay in our work differ from those in [1].
> >
> > To my understanding, both this paper and [1] use temporal decay to strengthen the training of earlier time-steps in the sampled trajectories.
> > In fact, statements similar to *emphasize the significance of earlier token (steps) in the sequence* appears several times in [1].
> > So I don't see major differences in the role and impact of temporal decay in this paper and [1].
> >
> > > 3. On the other hand, ours work is a good extension of [1].
> >
> > I think this statement ought to be clearly manifested in the paper, rather than keep saying things like "this paper proposes to integrate the temporal decay factor into DPO-family loss", which may not fairly reflect the methodological contribution of this paper and may overlay the contribution of [1].
> >
> > ### II. About the derivation/motivation
> >
> > > 4. By utilizing the DPO framework, .... eliminating issues related to delayed feedback.
> >
> > Not quite understand this claim, as DPO loss is derived under the bandit MDP assumption, which naturally, though implicitly, suffers from the delayed feedback issue due to the token-wise nature of LLM training losses and generation
> >
> > > 5. By accumulating the temporal decay reward of each token, we get the Eq.10
> >
> > I believe in Eq. (8), there should be a partition function (normalizer) depending on $s_t$ so that $\pi^*(\cdot \mid s_t)$ can integrate to $1$. How did those partition functions get cancelled in Eq.10?

---

> > > ### Author Response · Authors · 2024-11-26
> > > **Thanks for your response and we want to make the following revisions**
> > >
> > > I. About the novelty
> > > >> "Our research focuses on LLMs and their alignment was mainly conducted in text-to-image generation." "It is not easy to cover all papers in this kind of large research domains as there are so many papers published years."
> > > (1) These are not a strong-enough arguments given the similarity of the proposed loss with the listed prior work(s). (2) To my understanding, both this paper and [1] use temporal decay to strengthen the training of earlier time-steps in the sampled trajectories. In fact, statements similar to emphasize the significance of earlier token (steps) in the sequence appears several times in [1]. So I don't see major differences in the role and impact of temporal decay in this paper and [1].
> > >
> > > Thank you for your feedback regarding the novelty of our work. We would like to clarify the distinctions and contributions of our approach:
> > >
> > > - Different Perspectives and Tasks: While our work and the referenced prior work both involve preference optimization, they are derived from fundamentally different perspectives and are applied to different downstream tasks. The prior work focuses on text-to-image tasks, which involve fixed-length generation through a non-autoregressive diffusion process. In contrast, our research is centered on LLMs in an autoregressive context, where sequence generation dynamics are inherently different.
> > >
> > > - Motivation Differences: As illustrated in Figures 1 and 3 of our manuscript, our motivation diverges significantly from prior work. Our temporal decay mechanism is designed to address specific challenges in LLMs, such as length bias and the need for alignment with human preferences across varying sequence lengths.
> > >
> > > - Flexible Decay Mechanism: Our approach is not limited to exponential decay. As shown in Table 4, we explore multiple decay strategies, demonstrating the flexibility and adaptability of our method to different scenarios and tasks.
> > >
> > > - Theoretical Insights: We have provided a theoretical analysis based on the token-level MDP, suggesting the existence of an optimal gamma value for enhancing preference optimization. This theoretical foundation supports the practical effectiveness of our approach.
> > >
> > > - Extension and Complementarity: Our work serves as both an extension and a complement to the referenced study. While the prior work has not validated its approach on standard RLHF benchmarks, our method has been tested and shown to be effective in these contexts, as detailed in our experimental results.
> > >
> > > We have revised our manuscript to better highlight these distinctions (please refer to the introduction and method sections.) and have acknowledged the prior work as a significant related study that inspired our research. We hope these clarifications address your concerns and demonstrate the novelty and contribution of our work.
> > >
> > >
> > > > On the other hand, ours work is a good extension of [1].
> > > I think this statement ought to be clearly manifested in the paper, rather than keep saying things like "this paper proposes to integrate the temporal decay factor into DPO-family loss", which may not fairly reflect the methodological contribution of this paper and may overlay the contribution of [1].
> > >
> > > We have clarified the statement in Section 3.2 to accurately reflect the influence and methodological contribution of Work [1]. The revised text: "Inspired by the success of \citet{yang2024denseReward}, where earlier steps are crucial in the reverse chain of the diffusion denoising process, we propose a temporal decay mechanism to highlight the importance of earlier tokens in LLM scenarios." We believe this revision appropriately acknowledges the foundational work of \citet{yang2024denseReward} while clearly stating how our approach extends these ideas into the domain of large language models.

---

> > > ### Author Response · Authors · 2024-11-26
> > > **Towards the derivation**
> > >
> > > >> By utilizing the DPO framework, .... eliminating issues related to delayed feedback
> > > Not quite understand this claim, as DPO loss is derived under the bandit MDP assumption, which naturally, though implicitly, suffers from the delayed feedback issue due to the token-wise nature of LLM training losses and generation.
> > >
> > > A: The delayed feedback in traditional RL we emphasize here refers to a scenario where the consequences of an agent's actions are not immediately observed. The reward is only provided at the end of an episode and is zero at all other times, this is a typical case of delayed feedback known as sparse reward, as discussed in [2]. We mainly focus on DPO which is different from the traditional RL condition. In fact, although the original DPO is developed under the bandit assumption, it can also be derived within the token-level MDP setting and learn any dense reward function. Therefore, we claimed by utilizing the DPO framework, we avoid classical reinforcement learning altogether, eliminating issues related to delayed feedback.
> > >
> > > [2] Yang, Shentao, et al. "Preference-grounded token-level guidance for language model fine-tuning." Advances in Neural Information Processing Systems 36 (2023).
> > >
> > >
> > > >> By accumulating the temporal decay reward of each token, we get the Eq.10. I believe in Eq. (8), there should be a partition function (normalizer) depending on so that can integrate to 1. How did those partition functions get cancelled in Eq.10?
> > >
> > > A: From the soft Bellman equation interpretation of the maximum causal entropy distribution, the policy is distributed according to Eq.8, where $Q^*(s_t, a_t) \triangleq \beta \log Z_{a_t|s_t}, V^*(s_t) \triangleq \beta \log Z_{s_t}$[4]. Obviously, with $V^*(s_t) \triangleq \beta \log Z_{s_t}$, the policy $\pi^*$ integrates to 1. Although each step contains the state log partition function, we only use this expression to further simplify Eq.7(replace the item $Q^*(s_t, a_t) - V^*(s_t)$) and only $V^*(s_0)$ is remained in the Eq.10 which is similar to the partition function in the original DPO derivation. Assuming the chosen and rejected trajectory start at the same state $s_0$, the item $V^*(s_0) \triangleq \beta \log Z_{s_0}$ will cancel in the BT model. The derivation is different from [1], which does not consider the relation between reward and $Q^*$ under bellman equation.
> > >
> > > [4] Ziebart, Brian D. Modeling purposeful adaptive behavior with the principle of maximum causal entropy. Carnegie Mellon University, 2010.

---

> ### Comment · Reviewer_tbQd · 2024-11-26
> **Response to the authors**
>
> Dear authors,
>
> Thank you so much for the further revisions and clarifications, which are fair and clear explanations of the contribution and derivation of the proposed method. I've raised my ratings in account of these.
>
>
> ***
> A minor "errtum": IMHO diffusion reversed chain is also autoregressive similar to language generation, as ancestral sampling can be formulated as a first-order Markov process.

---

> > ### Author Response · Authors · 2024-11-27
> > **Thanks for the re-consideration!**
> >
> > Dear Reviewer tbQd
> >
> > Thank you very much for your response and for reconsidering your score. We will carefully review the details and further improve the manuscript to ensure the claims are precise and well-supported.

---

### Official Review · Reviewer_FaCH · 2024-11-03

**Soundness:** 2
**Presentation:** 2
**Contribution:** 2
**Rating:** 5
**Confidence:** 3

**Summary:**

The paper introduces a temporal decay mechanism that prioritizes earlier tokens in Direct Preference Optimization (DPO) to improve the alignment of large language models with human preferences. By dynamically adjusting the weights of tokens based on their position, the authors address the length bias seen in existing models, leading to enhanced learning from human feedback. Empirical results demonstrate that this approach outperforms traditional methods like SimPO and SamPO across benchmarks including AlpacaEval and Arena-Hard by large margin.

**Strengths:**

Presents significant advancements in preference optimization by linking temporal decay concepts to value learning within RL frameworks.

The authors conduct extensive, controlled experiments comparing their proposed method with a broad spectrum of established baselines, including SimPO, DPO, and KTO.

**Weaknesses:**

A main bottleneck of the paper is the reliance on automated benchmarks, which can be susceptible to manipulation or "hacking.", see [1]. This raises concerns about the validity of the results, suggesting that human evaluation is neccasary.

Additionally, the empirical results indicate that incorporating length control significantly increases the win rate of the proposed method. This finding suggests that it worth to further compare their approach with works [2] that also focus on length control and its impact on model performance.

[1] "Cheating automatic llm benchmarks: Null models achieve high win rates."
[2] "Meta-Rewarding Language Models: Self-Improving Alignment with LLM-as-a-Meta-Judge"

**Questions:**

According to some emprical observation, the gain in autobenchmark could come from length as well as better style control [3]. Have you tried testing on IFEval or gsm8k like tasks?

[3] "https://lmsys.org/blog/2024-08-28-style-control/"

---

> ### Author Response · Authors · 2024-11-18
> **Response to Reviewer FaCH**
>
> Thanks for your constructive feedbacks and we think all concerns would be well addressed in the folloing response!
>
> > A main bottleneck of the paper is the reliance on automated benchmarks, which can be susceptible to manipulation or "hacking.", see [1]. This raises concerns about the validity of the results, suggesting that human evaluation is neccasary.
>
> Thank you for bringing this important point to our attention and for referencing [1]. We have carefully reviewed the cited work and understand the potential risks associated with relying solely on automated benchmarks. Specifically, we recognize that some benchmarks might be vulnerable to manipulation, allowing models to artificially inflate performance without genuinely improving underlying capabilities.
>
> However, we believe that our results remain valid and robust for the following reasons:
>
> - We have evaluated our proposed method, D²PO, on a variety of widely used and rigorous benchmarks that are specifically designed to assess logical reasoning and problem-solving abilities. In addition to the initial automated benchmarks, we included GSM8K, MMLU, Math and OpenLLM Benchmark. The results are summarized in the General Response.
>
> - We have compared D²PO with previous methods, including DPO and SimPO, across all benchmarks. This comparative approach demonstrates that D²PO consistently achieves superior performance, indicating genuine improvements rather than artifacts of benchmark manipulation. Concretly, on both models (Llama3-8B and Gemma2-9B), D²PO demonstrates strong performance across all benchmarks. In the Gemma2-9B scenario, D²PO not only maintains performance but also surpasses the SFT baseline on TruthfulQA by 1 point. While other methods like DPO and SimPO exhibit significant performance degradation on several benchmarks (ARC-C, Hellaswag, TruthfulQA, and Winogrande), D²PO maintains or improves performance, suggesting that it enhances model ability without falling prey to benchmark-specific hacks.
>
> To further validate our results, we conducted human evaluations on the AlpacaEval2 and Arena-Hard datasets using the Gemma2-9B model. We enlisted four researchers who were not involved in this submission, with each person evaluating 50 samples for each benchmark. For each instruction, we randomized the order of the outputs from DPO and our proposed D²PO models to prevent bias. The evaluators assessed the responses based on three criteria: accuracy, completeness, and relevance, determining which response was better for each sample. If both responses were equally correct or incorrect, the result was considered a tie.
>
> The results of our comparison between D²PO and DPO are presented below. The findings indicate that D²PO achieved a significantly higher win rate than DPO, with a 67% win rate (computed by (win + tie/2) / total) overall in Arena-Hard and 69% in AlpacaEval 2. We hope this result can address your concern!
>
> | Method (Gemma2-9B)    | Win   | Tie   | Lose
> |-----------------------|-------|-------|-------
> | Arena-Hard            | 116   | 36    | 48
> | AlpacaEval 2          | 107   | 62    | 31
>
>
>
> > Additionally, the empirical results indicate that incorporating length control significantly increases the win rate of the proposed method. This finding suggests that it worth to further compare their approach with works [2] that also focus on length control and its impact on model performance.
>
>
> Thank you for your insightful comment and suggestion to compare the impact of length control on model performance between our work and the approach presented in [2]. We appreciate the opportunity to address this important aspect of our research.
>
> Reference [2] presents an innovative meta-rewarding mechanism for the self-improvement process, gradually enhancing the capabilities of both actors and judges through iterative methods. Their work focuses on self-play, aiming to improve the model's capabilities without relying on external models or human involvement, which is also a promising direction in the field. In contrast, our method primarily focuses on achieving better results compared to classical methods such as DPO under the same training data and hyperparameters. While [2] achieves length control mainly through data selection, our approach effectively reduces the length bias inherent in the dataset by employing a temporal decay strategy.
>
> Additionally, because [2] does not rely on external powerful reward models (RMs), the metrics on evaluation sets such as AlpacaEval2 are not as high. Furthermore, we cannot utilize the same training data as their method, making direct comparisons challenging. However, we believe that the meta-rewarding method proposed in [2] can be combined with our approach to replace external reward models, potentially enhancing performance without the need for external RMs. We will conduct this experiment as soon as we can acess their released-code.

---

> ### Author Response · Authors · 2024-11-26
> **Additional response to the question**
>
> > According to some emprical observation, the gain in autobenchmark could come from length as well as better style control [3]. Have you tried testing on IFEval or gsm8k like tasks?
>
> We have conducted comprehensive evaluations using the Arena-Hard benchmark, focusing on style control capabilities. Our results demonstrate that D²PO consistently achieves superior performance, even when style control is a key factor.
>
> Below are the results comparing D²PO with DPO and SimPO, highlighting our method's effectiveness in style-controlled settings:
>
>
> | Method (Gemma2-9B)    | AH   | AH(style control) |
> |-----------------------|------|-------------------|
> | D²PO (gamma0.98)      | 66.4 | 67.2              |
> | DPO (beta0.1)         | 56.7 | 57.2              |
> | DPO (beta0.01)        | 65.2 | 66.4              |
> | SimPO                 | 65.0 | 66.3              |
>
>
> We greatly appreciate it if you could take a moment to review our additional experiments and response and let us know if they adequately address your concerns!

---

> ### Author Response · Authors · 2024-11-27
> **We are looking forward to further discussion!**
>
> Dear Reviewer FaCH,
>
> We sincerely appreciate the time and effort you've dedicated to reviewing our paper. We understand that you have a busy schedule, and we are truly grateful for your valuable feedback. As the Author-Reviewer discussion phase approaches its end, we are eager to know whether our response has addressed your concerns and if there are any additional questions or points you'd like to discuss. We would greatly appreciate the opportunity to engage in further discussion if needed. Thank you once again for your thoughtful consideration.
>
> Best regards,
>
> All Authors

---

> ### Author Response · Authors · 2024-11-29
> **We apologize for any inconvenience and look forward to further discussion**
>
> Dear Reviewer FaCH,
>
> We apologize for bothering you again, but as the deadline approaches, we are eager to know whether our previous response has addressed your concerns. If your concerns have been adequately addressed, could you please reconsider your evaluation? We are also looking forward to discussing any additional questions or points you might have.
>
> Thank you once again for your time and thoughtful consideration.
>
> Best regards,
>
>  All Authors

---

> ### Author Response · Authors · 2024-12-02
> **Follow-Up on Our Rebuttal**
>
> Dear Reviewer FaCH
>
> We sincerely appreciate the time and effort you have dedicated to reviewing our paper. As the rebuttal period concludes tomorrow, we wanted to follow up to ensure that our responses have adequately addressed your concerns.
>
> - For the W1: We conducted a comprehensive human evaluation to confirm that our improvements are not a result of system manipulation. Additionally, we have included results from several benchmarks, including GSM8K, MMLU, Math, IFEval, and other open LLM benchmarks, as detailed in the general response table below for your convenience.
>
> | Method (Llama3-8B)    | MMLU  | GSM8K | Math  | IFEval prompt-strict | ARC-C (25) | Hellaswag(10) | TruthfulQA(0)| Winogrande(5)|
> |-----------------------|-------|-------|-------|----------------------|------------|---------------|--------------|--------------|
> | Instruct              | 61.66 | 78.47 | 7.91  | 68.58                | 61.95      | 78.80         | 51.62        | 75.53        |
> | D²PO (gamma0.98)      | 61.38 | 71.95 | 8.46  | 65.62                | 65.78      | 79.03         | 57.57        | 75.14        |
> | DPO                   | 56.66 | 70.51 | 7.77  | 65.06                | 65.10      | 79.99         | 56.38        | 74.51        |
> | SimPO                 | 55.22 | 57.54 | 5.27  | 60.81                | 67.58      | 78.82         | 63.83        | 74.27        |
>
>
> | Method (Gemma2-9B)    | MMLU  | GSM8K | Math  | IFEval prompt-strict | ARC-C (25) | Hellaswag(10) | TruthfulQA(0) | Winogrande(5)|
> |-----------------------|-------|-------|-------|----------------------|------------|---------------|---------------|--------------|
> | instruct              | 72.82 | 87.41 | 19.42 | 71.90                | 71.84      | 81.67         | 60.23         | 77.90        |
> | D²PO    (gamma0.98)   | 72.68 | 88.86 | 21.22 | 71.16                | 71.42      | 81.03         | 61.34         | 76.01        |
> | DPO                   | 72.21 | 88.53 | 19.42 | 60.07                | 69.88      | 71.48         | 57.65         | 72.69        |
> | SimPO                 | 72.35 | 88.17 | 19.00 | 71.53                | 68.34      | 66.51         | 58.87         | 73.72        |
>
> - We have compared our method with reference [2], recognizing that their approach is a self-improving method. Direct comparison is challenging due to the absence of their open-source code. It is worth noting that reference [2] is also submitted to ICLR 2025 concurrently, and we consider it as our concurrent work. We intend to explore this combination further in our next version as soon as its code becomes available.
>
> - We have conducted a thorough style control evaluation, and our method consistently outperforms existing approaches.
>
> We understand that you have a busy schedule, and we are truly grateful for your valuable feedback. Up to now, all three other reviewers have recognized the contributions of our paper and have updated their scores positively. We are fortunate to receive such supportive feedback and hope that our responses have satisfactorily addressed your concerns as well.
>
> As the Author-Reviewer discussion phase is nearing its end, we kindly request your reconsideration of your score based on the provided clarifications and additional results. If there are any further questions or points you would like to discuss, we are more than willing to engage in additional dialogue to ensure the quality of our work.
>
> We are looking forward to your response! Thanks for your time!
>
> Best regards
>
> All authors

---

### Official Review · Reviewer_ozx7 · 2024-11-04

**Soundness:** 2
**Presentation:** 3
**Contribution:** 2
**Rating:** 6
**Confidence:** 3

**Summary:**

The paper "Earlier Tokens Contribute More: Learning Direct Preference Optimization from Temporal Decay Perspective" introduces D2PO, a variant of Direct Preference Optimization (DPO) that uses a temporal decay factor to prioritize earlier tokens in model responses. By applying decay to later tokens, D2PO mitigates verbosity bias and aligns responses more closely with human preferences. Empirical results on benchmarks like AlpacaEval 2 and Arena-Hard demonstrate significant performance gains over standard DPO.

**Strengths:**

1. Innovative temporal decay mechanism that dynamically weights token importance, effectively addressing alignment biases.
2. Demonstrated gains over similar methods in benchmarks like AlpacaEval 2 and Arena-Hard, highlighting the model's capacity to generate concise and relevant responses.
3. Flexible design, capable of operating in both reference-based and reference-free settings.

**Weaknesses:**

1. Reliance on Instruction-Following Datasets and GPT-4 as the Judge Model: The results in this paper are primarily validated on instruction-following datasets, using GPT-4 as the judge model. These choices raise concerns about generalizability, as instruction-following datasets can introduce noise that may obscure certain aspects of model alignment. Additionally, relying solely on GPT-4’s judgments may bias evaluation outcomes. To ensure robust assessment, further evaluations on alternative, specialized datasets—such as math or logical reasoning datasets—would help verify D2PO's effectiveness in different contexts and task types. This additional testing would strengthen the validity of D2PO's benefits and clarify its performance across a broader range of alignment tasks.

2. Sensitivity to the Gamma Parameter: The performance of D2PO heavily relies on the gamma (γ) parameter, which controls the rate of decay across tokens. While gamma tuning allows flexibility, it may also necessitate extensive fine-tuning to achieve optimal performance across varied tasks. This dependency on precise gamma adjustment could make D2PO less practical in scenarios where rapid adaptation across diverse datasets is required. An exploration of adaptive or task-specific gamma values could improve generalizability and reduce the need for manual tuning.

**Questions:**

1. Would D2PO’s performance remain consistent on datasets that prioritize logical reasoning or mathematical tasks, where alignment needs may differ from instruction-following datasets?
2. Is there a specific range of gamma values that you would recommend based on your findings, or does it vary significantly across models?
3. Have you considered exploring an adaptive or task-specific gamma parameter to reduce sensitivity and improve D2PO’s generalizability?

---

> ### Author Response · Authors · 2024-11-18
> **Response to Reviewer ozx7 (W1 and W2)**
>
> Thanks for your constructive advice and we think all the conerns would be well addressed in our improved version. We would like to address your concerns as follows:
>
> > W1: Relying solely on GPT-4’s judgments may bias evaluation outcomes.
>
> Thank you for your insightful feedback. We acknowledge the importance of evaluating our method beyond instruction-following datasets to demonstrate its generalizability. To address this concern, we have conducted additional experiments on three widely used benchmarks focused on logical reasoning and mathematical problem-solving: MMLU, GSM8K, and Math. These benchmarks are designed to assess the reasoning abilities of large language models (LLMs) in diverse and complex tasks.
>
> The results of these experiments are presented in the General Response. comparing our proposed D²PO with the SFT models and the baselines DPO and SimPO. On the Llama3-8B configuration, our D²PO outperforms DPO and SimPO by a significant margin, particularly on the MMLU and Math benchmarks. Notably:
> - D²PO exhibits less performance degradation on GSM8K compared to SimPO, even though both methods effectively control output length.
> - D²PO achieves substantial performance gains on the Math dataset, surpassing the Instruct model by 0.55 points, while the other two methods show a noticeable decline.
>
> On the Gemma2-9B configuration, we observe a similar pattern, with D²PO demonstrating a significant performance advantage on the Math benchmark. These results suggest that D²PO effectively enhances reasoning and mathematical problem-solving abilities in LLMs across different models.
>
>
> These additional evaluations on specialized datasets confirm that D²PO maintains its effectiveness across different contexts and task types. By outperforming baseline methods on logical reasoning and math benchmarks, we strengthen the validity of D²PO's benefits and demonstrate its generalizability beyond instruction-following tasks.
>
>
> > W2: Sensitivity to the Gamma Parameter
>
> We appreciate your concern regarding the sensitivity of D²PO to the gamma ($\gamma$) parameter. In our study, we have already conducted ablation experiments to investigate the effect of varying γ values, as illustrated in Figure 5 of the paper. The findings indicate:
>
> - Consistency Across Models and Benchmarks: D²PO with a $\gamma$ slightly less than 1 delivers consistent performance gains across different models and benchmarks, including AlpacaEval 2, Arena-Hard, and MT-Bench.
> - Robustness to Gamma Variations: The performance of D²PO remains comparable across different $\gamma$ values in most scenarios, suggesting that the method is not overly sensitive to this parameter.
>
> In our experiments, we did not perform extensive tuning. All results presented in Tables 1 and 2 utilize the same $\gamma$ value of 0.98. Notably:
>
> - For models like Gemma2-9B and Mistral-12B, we used the same $\gamma$ value as Llama3-8B without additional tuning, and D²PO consistently outperformed DPO and SimPO.
> - Other methods, such as SimPO, also use specific hyperparameters detailed in their appendices, highlighting the generality and practicality of our approach.
>
> Our method's robustness to $\gamma$ suggests that it can be effectively applied without the need for exhaustive parameter tuning. This makes D²PO suitable for rapid adaptation across diverse datasets and reduces the need for manual hyperparameter adjustments.

---

> ### Author Response · Authors · 2024-11-18
> **Response to the Questions**
>
> > Q1: Would D²PO's performance remain consistent on datasets that prioritize logical reasoning or mathematical tasks?
>
> Yes, as we discussed in our response to Weakness 1, D²PO demonstrates strong performance on datasets that emphasize logical reasoning and mathematical problem-solving. Specifically, D²PO outperforms both DPO and SimPO on benchmarks like MMLU for logical reasoning and Math for mathematical tasks.
>
> Notably, D²PO achieves significantly higher performance than the SFT baseline on the Math benchmark—a result that the other two methods do not attain. A potential explanation for this improvement lies in our proposed temporal decay mechanism, which encourages the model to focus more on the initial part (prefix) of the generation. This emphasis on the prefix helps the model develop a clearer and more precise thought process, which is crucial for solving complex logical reasoning and mathematical problems. By enhancing the quality of the thought process, D²PO increases the likelihood of arriving at correct answers in tasks that require deep reasoning.
>
> > Q2: Is there a specific range of gamma values that you would recommend based on your findings, or does it vary significantly across models?
>
> Also a similar concern to Weakness 2. Based on our experimental results, we have found that setting the gamma value within the range of 0.97 to 0.98 is effective and consistent across most scenarios. Specifically:all experiments were conducted within a $\gamma=0.98$ in our main experiments. We have not specially tuned $\gamma$ for the purpose of generalizability, and we belive more specific tuned parameters would lead to a higher performance.
>
>
> > Q3: Have you considered exploring an adaptive or task-specific gamma parameter to reduce sensitivity and improve D2PO’s generalizability?
>
> Thank you for this insightful suggestion. Exploring an adaptive or task-specific gamma parameter is indeed an interesting direction for further enhancing D²PO's generalizability and performance. In our current work, we did not perform specialized tuning of the gamma parameter; instead, we used a general setting of $\gamma$ = 0.98 across all experiments. This value was chosen because it provided consistent performance improvements without the need for extensive tuning. We acknowledge that adapting the gamma parameter to specific tasks or models could potentially yield even better results than those we have reported. Here we report a $\gamma=0.99$ on Gemma2-9B. Here we found that both two models show consistent performance gains and $\gamma=0.99$ delivers better results in terms of IFEval and OpenLLM Benchmark. This results further demonstrate the effectiveness of our D²PO and we believe much finer adjustment of $\gamma$, e.g., 0.985 would be promising. While through the comparison of these two settings, we think task-specific gamma parameter indeed helps!
>
> | Method (Gemma2-9B)    | MMLU  | GSM8K | Math  | IFEval prompt-strict | ARC-C (25) | Hellaswag(10) | TruthfulQA(0) | Winogrande(5)|
> |-----------------------|-------|-------|-------|----------------------|------------|---------------|---------------|--------------|
> | instruct              | 72.82 | 87.41 | 19.42 | 71.90                | 71.84      | 81.67         | 60.23         | 77.90        |
> | D²PO    (gamma0.98)   | 72.68 | 88.86 | 21.22 | 71.16                | 71.42      | 81.03         | 61.34         | 76.01        |
> | D²PO    (gamma0.99)   | 72.43 | 88.32 | 21.08 | 71.72                | 72.35      | 81.63         | 62.76         | 76.40        |
> | DPO                   | 72.21 | 88.53 | 19.42 | 60.07                | 69.88      | 71.48         | 57.65         | 72.69        |
> | SimPO                 | 72.35 | 88.17 | 19.00 | 71.53                | 68.34      | 66.51         | 58.87         | 73.72        |

---

> ### Author Response · Authors · 2024-11-27
> **We are looking forward to further discussion!**
>
> Dear Reviewer ozx7,
>
> We sincerely appreciate the time and effort you've dedicated to reviewing our paper. We understand that you have a busy schedule, and we are truly grateful for your valuable feedback. As the Author-Reviewer discussion phase approaches its end, we are eager to know whether our response has addressed your concerns and if there are any additional questions or points you'd like to discuss. We would greatly appreciate the opportunity to engage in further discussion if needed. Thank you once again for your thoughtful consideration.
>
> Best regards,
>
>  All Authors

---

> > ### Comment · Reviewer_ozx7 · 2024-11-28
> >
> > Thank you for your response. It addressed my concerns. I will increase my rating to 6.

---

### Official Review · Reviewer_cJJC · 2024-11-12

**Soundness:** 3
**Presentation:** 3
**Contribution:** 3
**Rating:** 8
**Confidence:** 3

**Summary:**

This paper introduces D2PO (Decay-based Direct Preference Optimization), which aims to mitigate the length bias of DPO through a temporal decay parameter $\gamma$. This follows the principle that the quality of the overall sequence generated can be improved when the earlier tokens are more accurate. The authors show that D2PO outperforms baselines on AlpacaEval2, Arena-Hard and MT-Bench.

**Strengths:**

1. The authors motivate the problem clearly with the length bias problem of DPO with Figure 1, and showing that earlier tokens are more crucial in alignment settings. Figure 3 motivates the need for temporal decay instead of treating all tokens across a sequence uniformly.
1. Overall it is a complete paper which is well-written and easy to follow. As far as I can tell, the experimental results are also complete with thorough ablations.
1. The comparison between D2PO and the baselines (DPO, SimPO and SamPO) is very clear and I like the details in Figure 2 as well as Table 5 in the Appendix, in addition to the quantitative results.

**Weaknesses:**

1. Lack of Novelty: the derivation in 3.3 seems like a straightforward extension of Rafailov et, al. (2024) but using discounted return instead of return.
1. Lack of Theoretical Insight: Intuitively it makes sense that earlier tokens can be more important, and the authors do a good job providing experiments to support this. However, there is no formal motivation (e.g. some theoretical insights under the token MDP setting).

**Questions:**

1. In what kind of tasks or settings (e.g. types of datasets) do you think the assumption that earlier tokens are more important may not hold?

---

> ### Author Response · Authors · 2024-11-18
> **Response to Reviewer cJJC**
>
> Thanks for your useful feedback and we are willing to address your concerns as below:
>
> > W1: Lack of Novelty: the derivation in 3.3 seems like a straightforward extension of Rafailov et, al. (2024) but using discounted return instead of return.
>
> Thank you for your feedback. While our work builds upon Rafailov et al. (2024) [1], our key contribution is introducing a temporal decay mechanism in D²PO that assigns different weights to token-level rewards through discounting. This mechanism emphasizes earlier tokens in the sequence, aligning with Lin et al. (2024) [2]'s observation that initial tokens are more critical for alignment.
>
> Our motivation was to enhance DPO by integrating this temporal emphasis, resulting in practical benefits:
>
> - Length Control: D²PO achieves effective length control similar to methods like SimPO and SamPO, but with superior performance.
>
> - Improved Performance Across Benchmarks: We empirically demonstrate that D²PO outperforms existing methods on benchmarks such as AlpacaEval2, Arena-Hard, as well as on logical reasoning and mathematical tasks (Please refer to the General Response).
>
> In summary, while our derivation extends prior work, the practical advantages and empirical improvements offered by D²PO represent a novel and meaningful contribution to the field.
>
>
> [1] From r to Q∗: Your Language Model is Secretly a Q-Function. COLM 2024
>
> [2] The Unlocking Spell on Base LLMs: Rethinking Alignment via In-Context Learning. ICLR 2024
>
> > W2: Lack of Theoretical Insight: Intuitively it makes sense that earlier tokens can be more important, and the authors do a good job providing experiments to support this. However, there is no formal motivation (e.g. some theoretical insights under the token MDP setting).
>
> Thanks for your recognition of our empirical results. The observation that earlier tokens contribute more is motivated by the nature of the autoregressive process. To support our hypothesis, we further analyzed the token distribution shift of the alignment model, as shown in Figure 1, which indicates that the distribution of earlier tokens undergoes a significant change due to the alignment. Our contribution primarily lies in validating the effectiveness of this hypothesis. Additionally, this temporary decay method, owing to the favorable properties of convergent series (since $\sum \gamma^n$ has an upper bound), can also effectively reduce length bias. We have not yet reached a definite conclusion regarding a more in-depth theoretical analysis, but any future progress will be included in the manuscript promptly.
>
> > Q1: In what kind of tasks or settings (e.g. types of datasets) do you think the assumption that earlier tokens are more important may not hold?
>
> In tasks such as mathematical reasoning, obtaining high-quality training samples is particularly challenging. Many mathematical training samples are not easily accessible and are often constructed with the assistance of the LLM itself (We only have the right question and answer, but the CoT process is often obtained with the help of LLM). Currently, sample construction is based on standard answers, focusing primarily on whether the final answer is correct. However, in this context, the correctness of the reasoning process (the steps leading to the answer) is also critically important.
>
> It's possible for a response to contain incorrect reasoning steps yet arrive at the correct final answer. In such extreme cases, the earlier tokens, which represent the reasoning steps, cannot provide effective guidance. This limitation means that our method requires the training samples to be of very high quality. Otherwise, it may introduce noise into the preference optimization process.
>
> However, in our newly reported results on math and reasoning benchmarks (Please refer to the General Response), our D²PO method delivers better performance than both DPO and SimPO, even surpassing the SFT baseline on the Math benchmark. This indicates that when the reward model is precise enough, the temporal decay mechanism is beneficial for learning to solve such complex problems.

---

> ### Author Response · Authors · 2024-11-22
> **About the Weakness 2**
>
> Dear Reviewer cJJC,
>
> We appreciate your feedback. We have provided a theoretical analysis of our temporal-decay mechanism within the token MDP setting and have updated the manuscript accordingly (Section 4), highlighting revisions in blue. The detailed proof can be found in Appendix E.

---

> ### Author Response · Authors · 2024-12-01
> **Further discussion**
>
> Dear Reviewer cJJC,
>
> We sincerely appreciate the time and effort you've dedicated to reviewing our paper, and we thank you for your positive feedback on our work. To address your concerns, we have summarized the theoretical analysis in Section 4 for your convenience, believing that this analysis effectively responds to your points.
>
> Both DPO and our proposed method can be formulated as token-level Markov Decision Processes (MDPs). To understand the impact of the discount factor in this context, we conducted an in-depth analysis. We introduced the concept of suboptimality, which refers to the performance difference between our policy and the optimal policy, to quantify this impact. Our analysis derived an upper bound for suboptimality, consisting of two terms that vary monotonically with the discount factor $\gamma$ but in opposing directions:
>
> $\text{SubOpt}(\pi,s; \gamma_e) \leq 2(H - \frac{1-\gamma^H}{1-\gamma})R + \frac{2(1-\gamma^H)^2}{(1-\gamma)^2}E_{s \sim d^{\pi^{\*}}}\left[\mathbb{TV}(\pi^{\*}(a|s)||\pi(a|s)\right]R$,
>
> where $H$ denotes the finite horizon. The two terms in the upper bound reflect opposing influences on performance difference, highlighting the trade-offs involved. It indicates an optimal $\gamma^*$ where the balance between the opposing terms is achieved,  resulting in a minimized performance difference compared to the optimal policy. The detailed derivation can be found in Appendix E.
> Overall, in addition to introducing a temporal decay mechanism, our work provides a theoretical analysis that demonstrates the existence of an optimal discount factor, thereby enhancing the effectiveness of preference optimization in token-level MDPs.
>
> We understand that you have a busy schedule, and we are truly grateful for your valuable feedback. As the Author-Reviewer discussion phase approaches its end, we are eager to know whether our response has adequately addressed your concerns and if there are any additional questions or points you'd like to discuss. We would greatly appreciate the opportunity to engage in further discussion if needed.
>
> Thank you once again for your thoughtful consideration.
>
> Best wishes
>
> All Authors

---

> > ### Comment · Reviewer_cJJC · 2024-12-01
> >
> > Thank you for the clarifications and the effort during the rebuttal, especially with additional results on new benchmarks and theoretical analysis on the effect of the discount factor $\gamma$. I've updated the overall score from 6 to 8 and my confidence from 2 to 3. Overall, I think it is a solid paper that can benefit the research community.

---

### Author Response · Authors · 2024-11-18
**General Response for Additional Experiments**

To further substantiate our claims, we conducted additional experiments on three widely used benchmarks focused on logical reasoning and mathematical problem-solving: MMLU, GSM8K, and Math. These benchmarks are specifically designed to assess the reasoning abilities of LLMs in diverse and complex tasks. Additionally, we report results on the OpenLLM Benchmark to reinforce our assertion that our improvements do not come at the cost of general language modeling ability. Due to time constraints, our experiments were primarily conducted on the Llama3-8B and Gemma2-9B models.

| Method (Llama3-8B)    | MMLU  | GSM8K | Math  | IFEval prompt-strict | ARC-C (25) | Hellaswag(10) | TruthfulQA(0)| Winogrande(5)|
|-----------------------|-------|-------|-------|----------------------|------------|---------------|--------------|--------------|
| Instruct              | 61.66 | 78.47 | 7.91  | 68.58                | 61.95      | 78.80         | 51.62        | 75.53        |
| D²PO (gamma0.98)      | 61.38 | 71.95 | 8.46  | 65.62                | 65.78      | 79.03         | 57.57        | 75.14        |
| DPO                   | 56.66 | 70.51 | 7.77  | 65.06                | 65.10      | 79.99         | 56.38        | 74.51        |
| SimPO                 | 55.22 | 57.54 | 5.27  | 60.81                | 67.58      | 78.82         | 63.83        | 74.27        |


| Method (Gemma2-9B)    | MMLU  | GSM8K | Math  | IFEval prompt-strict | ARC-C (25) | Hellaswag(10) | TruthfulQA(0) | Winogrande(5)|
|-----------------------|-------|-------|-------|----------------------|------------|---------------|---------------|--------------|
| instruct              | 72.82 | 87.41 | 19.42 | 71.90                | 71.84      | 81.67         | 60.23         | 77.90        |
| D²PO    (gamma0.98)   | 72.68 | 88.86 | 21.22 | 71.16                | 71.42      | 81.03         | 61.34         | 76.01        |
| DPO                   | 72.21 | 88.53 | 19.42 | 60.07                | 69.88      | 71.48         | 57.65         | 72.69        |
| SimPO                 | 72.35 | 88.17 | 19.00 | 71.53                | 68.34      | 66.51         | 58.87         | 73.72        |

---

### Author Response · Authors · 2024-11-22
**Update the Manuscript**

Thank you once again for all the reviewers' efforts! We have updated our manuscript accordingly, and the revisions are summarized as follows:

- We conducted additional experiments on the OpenLLM benchmark, including MMLU, GSM8K, Math, and IFEval. Our results demonstrate that the improvements in RLHF benchmarks do not compromise general language modeling abilities.
- We performed human evaluations on both Arena-Hard and AlpacaEval2 datasets. The results indicate that our proposed D²PO significantly outperforms DPO in terms of the win rate.
- We provided a theoretical analysis explaining why our proposed temporal-decay mechanism is effective from the token MDP perspective. (Section 4)
- We have added the missing references raised by the reviewers.
- We corrected inaccurate statements to ensure clarity and precision.
- We moved the experimental setups (Appendix A) and some of the analyses (Appendix C) to the appendix.

We hope that these revisions address your concerns, and we look forward to discussing them with you. Since the discussion phase will be ending, we kindly request that you can read the response and make a reconsideration of our work. Please feel free to contact us if there are any points you would like to discuss further. Thank you for your valuable time!

---

### Author Response · Authors · 2024-11-26
**We are eager to discuss with dear reviewers**

Hi all reviewers,

Thank you once again for your valuable comments and suggestions, which have been very helpful to us. We understand this is a particularly busy time, and we greatly appreciate your efforts.

We kindly ask whether you could take a moment to review our additional experiments and responses to see if they adequately address your concerns. Should there be any further feedback, we are committed to incorporating it in the coming days.

Thank you for your time and consideration.

Best regards,

All Authors

---

### Meta-Review · Area_Chair_aauR · 2024-12-18

**Metareview:**

This paper proposes a token-level reward modeling method (R3HF) for reinforcement learning from human feedback, aiming to address limitations of existing RLHF approaches that use only a single, sequence-level reward. By treating reward prediction as a regression problem, the work was able to assign more granular token-level credit, thereby improving the learned policy.

Reviewers generally agree that the paper addresses a key limitation in current RLHF methods. Some reviewers initially questioned the novelty and technical insight from the work, but the concerns were generally addressed by additional discussions and additional experiments showing the improvements do not come at the cost of general language modeling ability. While gamma discounting of dense reward is a well studied concept in traditional RL, this work demonstrated that it has practical gains in RLHF of LLMs. I recommend accepting the paper.

**Additional Comments On Reviewer Discussion:**

The reviewers raised concerns around novelty of the proposed method given previous work on dense reward, a lack of theoretical insight, and sensitivity to the gamma parameter. The authors provided additional theoretical analysis and experiments demonstrating the gain of RLHF without sacrificing general language capabilities. One reviewer is particularly concerned with a lack of comparison to other methods that are also based on length control, but the author conducted additional experiments showing superior performance of the proposed method compared to length control method.

---

### Decision · Program_Chairs · 2025-01-22

Accept (Poster)